# Diet and Feeding Ecology of the Whitespotted Eagle Ray (*Aetobatus narinari*) from Florida Coastal Waters Revealed via DNA Barcoding

Brianna V. Cahill [1,*], Ryan J. Eckert [1], Kim Bassos-Hull [2], Thomas J. Ostendorf [1], Joshua D. Voss [1], Breanna C. DeGroot [1] and Matthew J. Ajemian [1]

[1] Harbor Branch Oceanographic Institute, Florida Atlantic University, 5600 US 1, Fort Pierce, FL 34946, USA; reckert2017@fau.edu (R.J.E.); tostendorf@fau.edu (T.J.O.); jvoss2@fau.edu (J.D.V.); bdegroot2017@fau.edu (B.C.D.); majemian@fau.edu (M.J.A.)
[2] Mote Marine Laboratory, 1600 Ken Thompson Pkwy, Sarasota, FL 34236, USA; kbhull@mote.org
[*] Correspondence: brianna.cahill@stonybrook.edu

**Abstract:** The whitespotted eagle ray (*Aetobatus narinari*) is a highly mobile, predatory batoid distributed throughout shallow, warm–temperate to tropical Atlantic Basin waters from North Carolina to Brazil. The species' strong, plate-like dentition facilitates the consumption of hard-shelled prey, and due to effective winnowing behavior, it is a significant challenge to identify prey based on soft tissues alone. Here, we report on the first analysis of whitespotted eagle ray diet in Florida waters using visual-based gut content analysis complemented with DNA barcoding. Gut contents were obtained via gastric lavage from 50 individuals collected in the Indian River Lagoon and off Sarasota, Florida. Of the 211 unique prey samples collected, 167 were deemed suitable for sequencing. Approximately 56.3% of samples yielded positive species matches in genetic databases. Results from the sequenced data indicate that the whitespotted eagle ray diet in Florida is mainly comprised of bivalves and gastropods, with variable inclusion of crustaceans. Despite positive identification of venerid clams, there was no evidence for the consumption of hard clams (*Mercenaria* spp.), a major shellfish aquaculture and restoration species in Florida. Such wide-ranging prey species from various trophic guilds and locations highlight the whitespotted eagle ray's diverse role in the top-down regulation of coastal benthic communities.

**Keywords:** batoid; invertivore; gastric lavage; durophagy

**Key Contribution:** Using DNA barcoding coupled with a traditional visual description of diet, we found that *A. narinari* exhibited a highly variable, locale-dependent diet consisting of bivalves, gastropods, and crustaceans. Though no common commercially important bivalves were observed in the gut contents, predatory gastropods that commonly feed on these commercially important species were identified, emphasizing the variable role *A. narinari* plays in structuring benthic mollusk communities through direct and indirect consumptive effects.

## 1. Introduction

Batoids, including skates and rays, are a diverse grouping of cartilaginous fishes that comprise over 630 species within 26 families across four orders [1]. While most batoids are benthic-dwelling and are heavily reliant on a benthic habitat [2], there are four families that are considered pelagic or benthopelagic: Aetobatidae, Rhinopteridae, Mobulidae and Myliobatidae [1]. Rays within these four families can make large-scale movements and therefore exist in a wide variety of environments. For example, whitespotted eagle rays (*Aetobatus narinari*) are highly mobile rays common in the western Atlantic, including the Gulf of Mexico [3] which can dive to depths of 50.5 m but primarily occupy the upper 10 m of the water column [4] and reach disc widths of 2 m [5]. The International Union for

Conservation of Nature (IUCN) Red List of Threatened Species classifies whitespotted eagle rays as "Endangered" [6] with a decreasing trend in population, though both Florida and Alabama have implemented protections for whitespotted eagle rays, preventing targeted fisheries and harvest. Despite their protected status, there is limited information available on critical components of their life history, such as diet.

Whitespotted eagle ray diet has been assessed in other regions throughout their western Atlantic range. At the northern extent of their distribution, in North Carolina, their diet was described to consist exclusively of hard clams (*Mercenaria mercenaria*) [7], although this assessment was qualitative. Additionally, these initial reports of whitespotted eagle rays described their unique jaw morphology, consisting of hard plate-like dentition, with a spade-shaped bottom plate to facilitate shell-crushing behaviors [7–9], and described finding stomach contents devoid of identifiable shell pieces. Later, studies of the whitespotted eagle ray diet from the greater Caribbean region suggested diets consisting of conch species, including queen (*Strombus gigas*) and rooster conch (*Strombus gallus* [10,11]), which similarly were devoid of shell and identifiable opercula. Gut contents of whitespotted eagle rays sampled in Bermuda consisted primarily of bivalves, including calico clams (*Macrocallista maculata*), lucinid clams (*Codakia* sp.), eared arks (*Anadara notabilis*) and purplish tagelus (*Tagelus divisus*), along with a few gastropods such as milk conch (*Strombus costatus* [12]). Conversely, whitespotted eagle ray diets analyzed in Mexico consisted primarily of gastropods, including the West Indian fighting conch (*Strombus pugilis*), netted olive snail (*Americoliva reticularis*) and milk conch (*Lobatus costatus*), in addition to the giant red hermit crab (*Petrochirus diogenes*) for large females [13]. Finally, congeners (i.e., *Aetobatus ocellatus*) caught in the Indo-Pacific had diets that consisted primarily of gastropods, with a moderate importance of bivalves in both Australia and Taiwan [14]. Additionally, the diet for individuals caught in Australia also included crustaceans [14]. In most aforementioned cases (excepting [12]), rays were sacrificed, and the entire digestive tract was assessed; however, all previous cases used visual identification to describe the diet.

Visual assessment of gut contents can facilitate viable prey identification [15]; however, this approach can benefit substantially from supplemental information provided by DNA barcoding techniques, particularly when prey items are partially digested and/or unrecognizable [16]. DNA barcoding is a tool used to rapidly identify species using small regions of the genome which possess species-level genetic variation positioned between conserved flanking regions [17]. Due to the high variability within specified genes (typically 300–600 base pairs), barcoding provides an ideal opportunity to differentiate species while avoiding the limitations of physical identification conducted historically [18]. The cytochrome *c* oxidase subunit 1 gene (CO1) is a well-known genetic marker for animals [19], making it an ideal gene to utilize for the DNA barcoding of whitespotted eagle ray gut contents, which are generally devoid of accompanying mollusk shells.

In addition to filling general ecological knowledge gaps for whitespotted eagle rays, these dietary data can also provide insight into this species' potential to interact with shellfish enhancement activities (i.e., bivalve aquaculture and restoration) that occur in Florida coastal waters. Due to hard clam stock collapses during the 1990s, caused by overharvest and suboptimal environmental conditions, ambient hard clam populations remain depleted in this region [20]. Since then, 1500 underwater acres have been designated to serve as locations for the "grow-out" of hard clams produced by shellfish aquaculture [21,22], and even more acreage has been designated for shellfish restoration. Specifically, in the Indian River Lagoon, Florida, durophagous rays (e.g., whitespotted eagle rays and cownose rays (*Rhinoptera bonasus*)) have been perceived as threats to enhancement activities by clam farmers (E. Mangano pers. comm.; Ajemian, unpublished data). Previous research suggests that whitespotted eagle rays can consume clams and manipulate gear commonly used in hard clam aquaculture [23] and that the rays can spend extensive periods of time within range of clam leases [24]. Unfortunately, there are no data to support whether rays are indeed depredating clams from these areas or consuming other associated fauna. By assessing the

diets of rays caught in proximity to shellfish enhancement locations, we can potentially determine whether these predators are provisioned by these operations. Here, we used a combination of DNA barcoding techniques and visual identification to describe the diet of whitespotted eagle rays from Florida coastal waters using non-lethal collection techniques. We hypothesized that whitespotted eagle ray diets would (1) differ between coastlines (i.e., Atlantic vs. Gulf), (2) be comparable among locations considered on the Atlantic coast, (3) vary across ontogeny, and that (4) commercially important bivalves would not comprise a majority of the diet.

## 2. Materials and Methods

### 2.1. Invertebrate Collection

To facilitate visual prey identification, invertebrate species were opportunistically collected by hand and haphazard sieving was conducted on accessible sandbars and in shallow (<1.5 m), benthic habitats in four primary locales. These included sites surrounding three major inlets on the Atlantic Coast (Fort Pierce, Sebastian, St. Lucie) and two from the Gulf coast (New Pass and Big Pass) off Sarasota, Florida. All invertebrates were stored in a −20 °C chest freezer until post-processing. During post-processing, all individuals were identified visually, and internal tissue was photographed to create an invertebrate tissue identification guide for the processing of unknown remains from extracted gut contents.

### 2.2. Gut Content Collection

Whitespotted eagle rays were targeted in the same four locations, including Fort Pierce, Sebastian, St. Lucie and Sarasota, FL, and were caught using (a) a 200 × 3 m tangle net, (b) a 500 × 3 m nylon seine net or (c) a 200 × 4 m knotted tangle net (Figure 1). Once trapped within an enclosed compass, the ray was moved to an onboard livewell, supplied with free-flowing oxygenated water. Each individual received a full workup involving full body and buccal measurements (including dental plate width (cm) and gape (cm)), biological sample collection, external tagging and pulsed gastric lavage. Gastric lavage was conducted using a bilge pump (Rule 360 GPH 24DA Standard, Xylem Inc., Beverly, MA, USA), submerged in ambient seawater, connected to 3 m of 9.5 mm, 12.7 mm or 15.8 mm diameter polyester reinforced clear PVC tubing (Shields Rubber Co., Erie, PA, USA), depending on the size of the animal. Rays designated as young of year (<70 cm disc width [25], hereafter referred to as YOY), received the smallest diameter hose of 9.5 mm; immature rays (70–127 cm) received the medium diameter hose of 12.7 mm; mature rays (>127 cm [5]) received the largest diameter hose of 15.8 mm.

Once positioned, a fine-mesh produce bag (Ecowaare, approximately 0.5 × 0.5 mm) was secured around the pelvic fins and cinched tight with the drawstring, anterior to the cloaca, and the ray was tilted so the posterior portion of the body was lower than the anterior. The gastric lavage tube was then fed into the mouth, down the esophagus past the pectoral girdle, and into the stomach where ambient seawater flushed the ray's digestive tract for up to two minutes or until clear seawater was flowing from the cloaca following the collection of the contents (Figure 2). Once the gastric lavage procedure was complete, the mesh bag was transferred to a cooler and stored on ice. At the lab, mesh bags containing the gut contents were transferred to a pre-weighed 405 × 405 mm mesh piece (BetterVue 1 × 1 mm, 48″ × 25′ roll, Phifer, Tuscaloosa, AL, USA) for long-term storage in a −20 °C chest freezer until processing. During processing, gut contents were thawed at room temperature, sorted into like species, photographed, then enumerated and weighed (0.001 g). Subsamples of like species, hereafter referred to as gastric lavage samples, were cut into 2 × 2 mm pieces, placed into 2 mL cryovials with 70% ethanol and stored in the −20 °C freezer until DNA extraction.

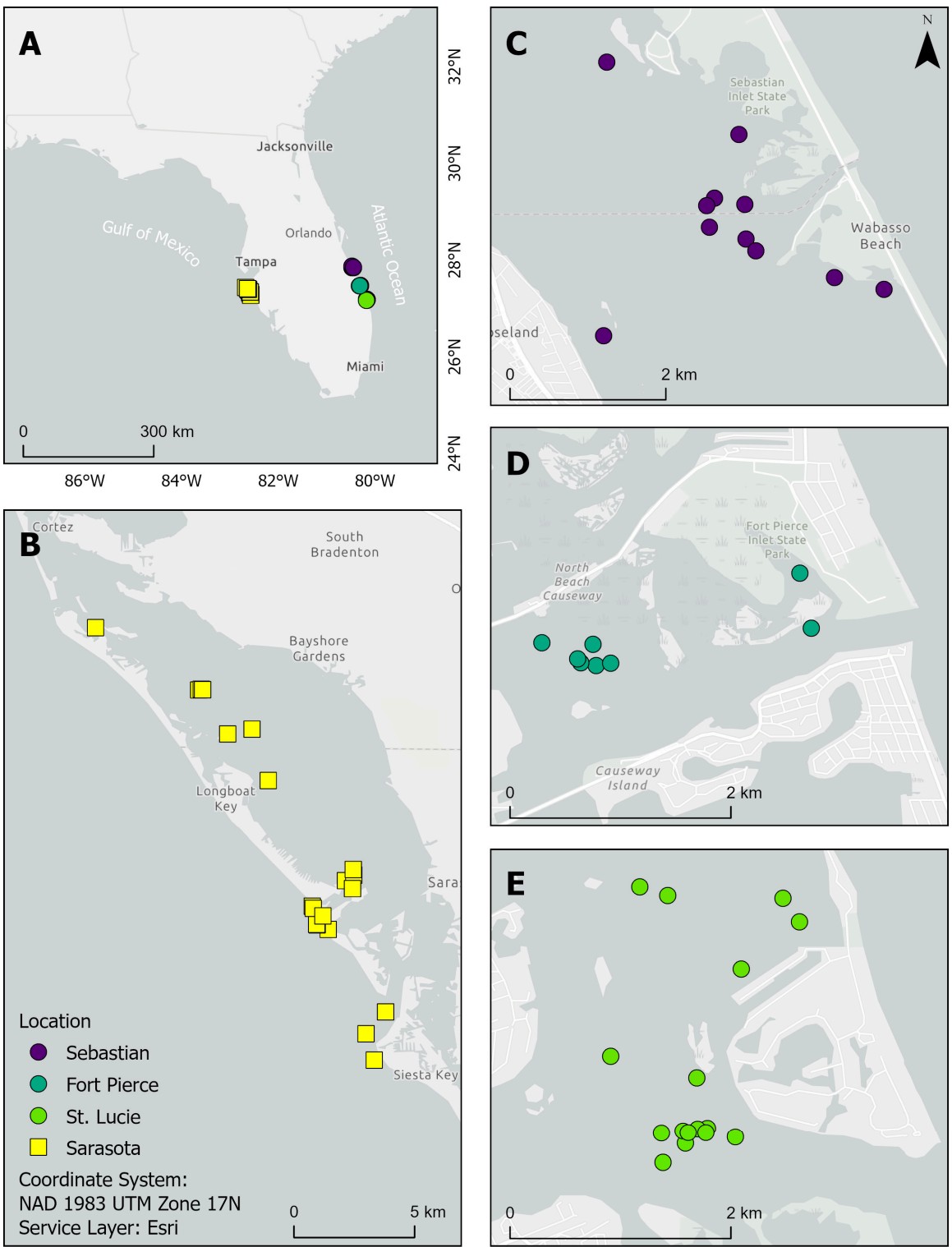

**Figure 1.** Whitespotted eagle ray sampling locations showing (**A**) three locations on the Atlantic Ocean coastline and one on the Gulf of Mexico, and capture locations in (**B**) Sarasota, (**C**) Sebastian, (**D**) Fort Pierce and (**E**) St. Lucie. These are the capture locations for all individuals.

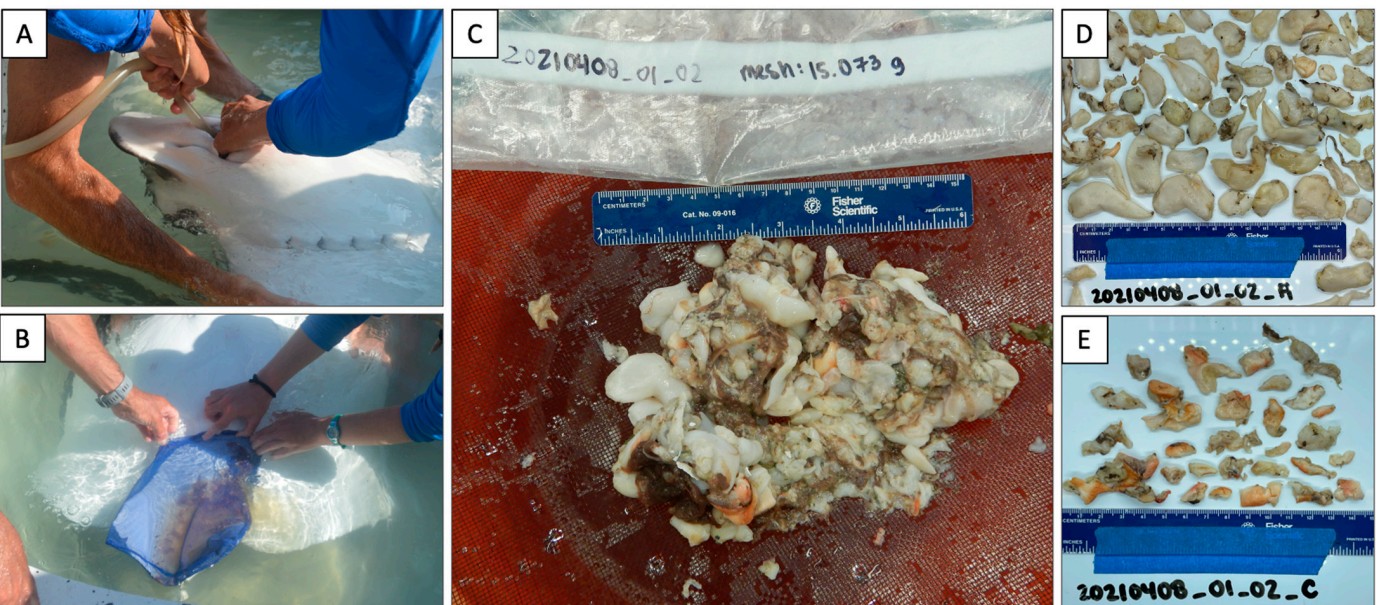

**Figure 2.** Photos depicting sampling process including (**A**) the insertion of the tube for pulsed gastric lavage, (**B**) the placement of the mesh produce bag surrounding the cloaca to collect gut contents, (**C**) a set of gut contents once migrated to the mesh and (**D,E**) subsamples from the same set of gut contents.

### 2.3. DNA Barcoding Preparation

Prior to extraction, approximately ~30 mg of tissue from each subsample was placed in 500 μL of TRIzol for a minimum of 30 min. Tissue was then transferred into 2 mL FastPrep tubes with 1000 μL of dispersion buffer with proteinase K, RNAse A, and 3–4 2.5 mm glass beads for maceration using three cycles of 45 s intervals with a bead homogenizer (MP FastPrep 24, MP Biomedicals, Irvine, CA, USA). Samples were then incubated at 55 °C overnight while mixing. Following the overnight proteinase K digestion, 150 μL of potassium acetate (KOAc, 5 M) was added to each tube, inverted, incubated at 60 °C for 10 min, and then centrifuged at 20,000× $g$ for 15 min. Samples were then centrifuged again for 3 min at 20,000× $g$ and 4 °C to pellet the beads and debris. DNA was then extracted using a modified phenol-chloroform-isoamyl alcohol extraction [26]. Samples were then cleaned with the Zymo DNA Clean and Concentrator-5 kit (Irvine, CA, USA) following the manufacturer's protocol and eluted with nuclease-free water. The quality of extracted DNA samples was confirmed using a NanoDrop 2000 (Thermo Fisher, Waltham, MA, USA) for 260/280 and 260/230 values, and DNA was quantified using a Qubit 4.0 fluorometer (Thermo Fisher, Waltham, MA, USA). DNA was diluted to a concentration of 2 ng μL$^{-1}$.

All successfully extracted samples were amplified using the redesigned COI primers jgLCO1490 (5′-TIT CIA CIA AYC AYA ARG AYA TTG G-3′) and jgHCO2198 (5′-TAI ACY TCI GGR TGI CCR AAR AAY CA-3′ [27]). Each 20 μL polymerase chain reaction (PCR) consisted of 4 ng of DNA template, 0.65U Taq polymerase (Takara Ex Taq, Takara Bio USA, Inc., San Jose, CA, USA), 1.5X Taq buffer, 0.235 mM deoxy-nucleoside triphosphate (dNTP) and 0.5 μM each primer. Negative controls were included in every PCR reaction to assess for contamination. PCR cycling conditions followed a 3-step touchdown protocol consisting of an initial denaturation at 98 °C for 10 s. Cycle profiles (n = 35) were as follows: 10 s of denaturation at 98 °C, followed by 30 s of annealing, and 60 s of extension at 72 °C. Annealing temperatures began at 50 °C and decreased by 1 °C for each subsequent cycle until reaching a final annealing temperature of 48 °C. The touchdown protocol finished with a final extension of 72 °C for five minutes, and then held at 4 °C. PCR product was confirmed using electrophoresis on a 2% agarose gel stained with ethidium bromide.

Sequencing was conducted on an Applied Biosystems 3730XL DNA Analyzer by the University of Arizona Genetics Core.

Contig sequence results were generated, when possible, by the University of Arizona Genetics Core using Geneious Prime (Version 2023.2). When contigs could not be created, forward and reverse sequences were provided, unless there were quality issues. All sequence data were assessed using the Basic Local Alignment Search Tool in the NCBI GenBank (http://www.ncbi.nlm.nih.gov/, accessed on 17 January 2023) and Barcode of Life databases (http://www.boldsystems.org/index.php/, accessed on 17 January 2023) for species identification. Samples with high (>95%) sequence matches to information available in either database were identified as such, and samples with low (<95%) sequence matches were identified visually using the invertebrate tissue guide and gastric lavage sample photos. Photos of samples with high sequence matches were sorted first, and samples with low sequence matches were sorted afterwards using tissue shape, color and other defining characteristics by two people and a consensus was derived. Though infrequent, misidentified tissues from gut contents were discovered during the visual classification process and corrected (Supplementary Document S1). The correction process involved generating an estimated weight from other gastric lavage samples of the same species identification and size and subtracting that estimated weight from the original sample.

### *2.4. Data Analyses*

Data were imported into the statistical software R (Version 4.0.4) for analysis. For each individual, the total weight of the gut contents was divided by the weight of the ray to determine the proportion of total body weight. Vacuity, or the proportion of individuals without gut contents to total individuals sampled, was assessed by location and life stage. Cumulative prey curves were generated using the *vegan* package [28] to assess the total number of unique prey items collected from all rays sampled [13,29,30]. Using these data, multiple model structures (e.g., bootstrap, chao, jackknife 1, jackknife 2) were fit, and a linear regression was fit to the last four points of the modeled sample curve to determine sample size sufficiency [31]. Familial presence in gut contents was assessed using frequency of occurrence (%FO), consisting of the frequency of that family observed for all individuals within each location. Using the total weights, proportion of diet by class was assessed by life stage and capture location. Additionally, the index of importance (%IOI) for each prey species observed was calculated using the overall percentage weight (%W) and %FO [32].

Using the weights of each identified prey item in the gut contents, we tested the null hypothesis that whitespotted eagle ray dietary composition did not vary by space or ontogeny. After standardization to the individual's weight, taxon-specific contributions (i.e., variables) were first square-root-transformed, and then used to build a Bray–Curtis dissimilarity matrix, using individual rays as replicates (samples). The matrix was then ordinated via non-metric multidimensional scaling using the *vegan* package [28] to assess the relationships between samples by life stage and capture location. Non-metric dimensional scaling plots (nMDS) were selected as they are a non-parametric ordination approach to assess community structure [33]. A permutational multivariate analysis of variance (PERMANOVA) was used to assess the combined effects of location and maturity state (fixed factors) on prey composition and contribution to the diet. Multilevel pairwise PERMANOVAs were conducted using the *pairwiseAdonis* package [34]. Similarity percentage (SIMPER) analysis was conducted to identify the species most greatly contributing to the dissimilarity for significant pairwise comparisons. The threshold for significance was defined as $\alpha = 0.05$.

### 3. Results

Overall, there were 61 rays caught and lavaged in this study, but only 50 yielded gut contents (Table 1). The disc width of the sampled animals ranged from 66.8 to 182.2 cm (131.9 ± 9.2 cm; n = 10) in Sebastian, 125.0 to 187.4 cm (142.5 ± 8.1 cm; n = 8) in Fort Pierce, 99.2 to 166.0 cm (134.1 ± 3.4 cm; n = 16) in St. Lucie and 54.0 to 184.2 cm (123.7 ± 10.0 cm;

n = 17) in Sarasota. Buccal measurements, including plate (upper and bottom) and gape, increased isometrically with whitespotted eagle ray capture size (Figure 3). Gut content weight ranged from 0.576 to 150.190 g among individuals. Vacuity was variable among locations, accounting for 9.1% of rays in Sebastian, 11.1% in Fort Pierce, 11.8% in St. Lucie and 29.2% in Sarasota (Table 2). Vacuity was 25% for YOY, 9.5% for juvenile and 22.2% for mature rays. We only captured three YOY (<70 cm, n = 3) with gut contents, thus all immature rays (<127 cm) were grouped for statistical analyses. The number of unique taxa observed for each individual ranged from 1 to 7 items, but overall, there were 33 unique prey items observed in the gut contents. Using these data, we fitted cumulative prey curves to the Atlantic and Gulf coast; however, after fitting multiple model structures to our data, the linear regression suggested an insufficient sample size for the Atlantic ($R^2 = 0.991$; $p = 0.003$; S = 31; Figure 4) and Gulf coast ($R^2 = 0.996$; $p = 0.001$; S = 11).

**Table 1.** Information for all whitespotted eagle rays (SER) with gut contents. Disc width (DW) is shown in cm, whitespotted eagle ray weight is in kg and gut content weight (GC Weight) in g refers to the total weight of all tissues present in the gut contents. Unique taxa refer to the number of unique prey items observed in the gut contents for each ray.

| ID | Date | Coast | Location | Sex | Life Stage | DW (cm) | SER Weight (kg) | GC Weight (g) | Unique Taxa |
|---|---|---|---|---|---|---|---|---|---|
| SER1 | 22 May 2020 | Gulf | Sarasota | F | Mature | 165.4 | 66.6 | 14.3 | 2 |
| SER2 | 27 May 2020 | Atlantic | Fort Pierce | F | Mature | 135.6 | 35.9 | 0.6 | 2 |
| SER3 | 28 May 2020 | Gulf | Sarasota | F | Mature | 157.8 | 64.4 | 22.8 | 1 |
| SER4 | 28 May 2020 | Gulf | Sarasota | F | Mature | 184.2 | 91.2 | 18.6 | 4 |
| SER5 | 29 May 2020 | Gulf | Sarasota | F | Mature | 179.0 | 96.1 | 7.4 | 2 |
| SER6 | 29 May 2020 | Gulf | Sarasota | F | Mature | 164.0 | 73.0 | 2.2 | 1 |
| SER7 | 9 June 2020 | Atlantic | Sebastian | M | Mature | 134.6 | 32.3 | 18.2 | 3 |
| SER8 | 22 June 2020 | Atlantic | Sebastian | M | Mature | 141.8 | 42.1 | 3.2 | 3 |
| SER9 | 23 June 2020 | Atlantic | St. Lucie | M | Juvenile | 120.8 | 26.1 | 8.2 | 4 |
| SER10 | 9 July 2020 | Atlantic | Fort Pierce | F | Juvenile | 126.2 | 30 | 115.5 | 6 |
| SER11 | 9 July 2020 | Atlantic | Fort Pierce | F | Juvenile | 125.0 | 27.9 | 14.9 | 4 |
| SER12 | 21 July 2020 | Atlantic | Sebastian | M | Mature | 133.5 | 37.2 | 20.2 | 3 |
| SER13 | 6 August 2020 | Atlantic | Sebastian | M | Juvenile | 126.0 | 30.3 | 9.5 | 7 |
| SER14 | 6 August 2020 | Atlantic | Sebastian | M | Mature | 128.0 | 31.6 | 8.3 | 2 |
| SER15 | 20 August 2020 | Gulf | Sarasota | M | Juvenile | 114.8 | 18.9 | 0.7 | 2 |
| SER16 | 28 September 2020 | Gulf | Sarasota | M | Juvenile | 106.2 | 20.2 | 8.1 | 1 |
| SER17 | 28 September 2020 | Gulf | Sarasota | M | Juvenile | 112.0 | 19.3 | 9.4 | 2 |
| SER18 | 26 October 2020 | Gulf | Sarasota | F | Juvenile | 111.2 | 22.0 | 1.5 | 1 |
| SER19 | 16 December 2020 | Atlantic | Sebastian | M | YOY | 66.8 | 4.1 | 9.2 | 4 |
| SER20 | 16 December 2020 | Atlantic | Sebastian | F | Mature | 182.2 | 94.1 | 15.2 | 2 |
| SER21 | 17 December 2020 | Atlantic | Fort Pierce | F | Mature | 187.4 | 102.4 | 40.4 | 4 |
| SER22 | 17 December 2020 | Atlantic | Fort Pierce | F | Mature | 172.4 | 79.6 | 18.2 | 3 |
| SER23 | 16 February 2021 | Atlantic | Fort Pierce | M | Mature | 128.6 | 31.3 | 6.9 | 3 |
| SER24 | 17 February 2021 | Atlantic | St. Lucie | M | Mature | 140.8 | 36.1 | 11.4 | 6 |
| SER25 | 17 February 2021 | Atlantic | St. Lucie | M | Mature | 137.6 | 37.6 | 36.8 | 5 |
| SER26 | 17 February 2021 | Atlantic | St. Lucie | M | Mature | 138.4 | 36.8 | 26.3 | 4 |
| SER27 | 17 February 2021 | Atlantic | St. Lucie | M | Juvenile | 124.6 | 25.7 | 26.8 | 5 |
| SER28 | 17 March 2021 | Atlantic | St. Lucie | M | Mature | 133.4 | 36.4 | 23.0 | 2 |
| SER29 | 17 March 2021 | Atlantic | St. Lucie | F | Juvenile | 123.2 | 27.4 | 11.7 | 1 |
| SER30 | 18 March 2021 | Atlantic | St. Lucie | M | Mature | 143.8 | 43.7 | 8.0 | 3 |
| SER31 | 18 March 2021 | Atlantic | St. Lucie | F | Juvenile | 123.0 | 29.6 | 8.7 | 1 |
| SER32 | 18 March 2021 | Atlantic | St. Lucie | F | Mature | 166.0 | 68.9 | 30.1 | 4 |
| SER33 | 29 March 2021 | Atlantic | Sebastian | M | Mature | 139.0 | 39.4 | 105.0 | 4 |
| SER34 | 29 March 2021 | Atlantic | Sebastian | M | Juvenile | 110.8 | 18.6 | 6.8 | 2 |
| SER35 | 7 April 2021 | Gulf | Sarasota | F | Juvenile | 122.0 | 27.9 | 11.1 | 1 |
| SER36 | 8 April 2021 | Gulf | Sarasota | F | Juvenile | 118.0 | 26.5 | 3.7 | 1 |
| SER37 | 8 April 2021 | Gulf | Sarasota | F | Mature | 137.2 | 38.9 | 150.2 | 3 |
| SER38 | 8 April 2021 | Gulf | Sarasota | M | Juvenile | 105.4 | 17.8 | 54.1 | 2 |
| SER39 | 22 April 2021 | Gulf | Sarasota | F | Juvenile | 84.9 | 8.9 | 16.1 | 2 |
| SER40 | 22 April 2021 | Gulf | Sarasota | M | Juvenile | 124.5 | 31.7 | 67.4 | 2 |
| SER41 | 22 April 2021 | Gulf | Sarasota | F | YOY | 54.9 | 2.5 | 11.4 | 2 |
| SER42 | 23 April 2021 | Gulf | Sarasota | F | YOY | 54.0 | 2.5 | 9.5 | 3 |
| SER43 | 13 July 2021 | Atlantic | Sebastian | F | Mature | 165.8 | 69.25 | 32.6 | 4 |
| SER44 | 15 July 2021 | Atlantic | St. Lucie | M | Mature | 128.8 | 41.2 | 12.1 | 5 |
| SER45 | 15 July 2021 | Atlantic | St. Lucie | M | Mature | 149.8 | 50.3 | 95.4 | 4 |
| SER46 | 15 July 2021 | Atlantic | St. Lucie | F | Mature | 140.0 | 41.15 | 4.9 | 1 |
| SER47 | 15 July 2021 | Atlantic | St. Lucie | F | Juvenile | 112.2 | 22.8 | 5.6 | 3 |
| SER48 | 15 July 2021 | Atlantic | St. Lucie | M | Juvenile | 99.2 | 13.5 | 2.8 | 1 |
| SER49 | 25 August 2021 | Atlantic | Fort Pierce | F | Mature | 160.4 | 59.5 | 20.5 | 3 |
| SER50 | 25 August 2021 | Atlantic | Fort Pierce | F | Mature | 132.8 | 35.6 | 21.6 | 7 |

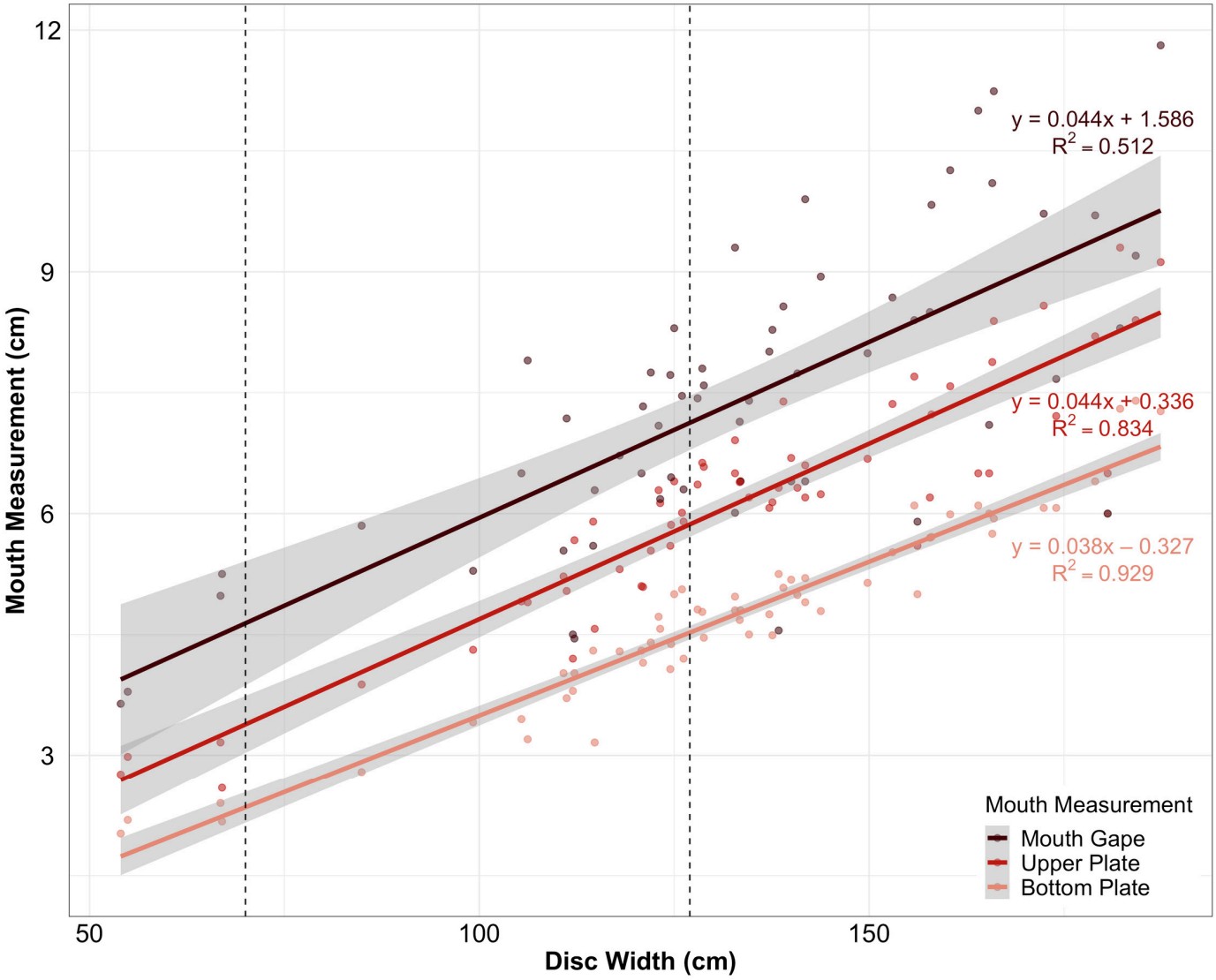

**Figure 3.** Mouth measurements for all whitespotted eagle rays captured over the duration of the study. The dotted lines refer to breaks in animal life stage denoting young of year (YOY, <70 cm), juveniles (70–127 cm) and mature individuals (>127 cm). A linear regression was fit to each mouth measurement.

**Table 2.** Collection information with the number of individuals sampled at each location and life stage. Total refers to the total number of animals caught and sampled but includes individuals that did not have gut contents. Number of rays without gut contents is shown within the square brackets ([ ]) for each location and life stage. The % Vacuity refers to the proportion of the total where animals did not have gut contents.

| Life Stage | Sebastian | Fort Pierce | St. Lucie | Sarasota | Total | % Vacuity |
|---|---|---|---|---|---|---|
| YOY | 1 | 0 | 0 | 2 | 4 [1] | 25.0% |
| Juvenile | 2 | 2 | 6 | 9 | 21 [2] | 9.5% |
| Mature | 7 | 6 | 10 | 6 | 36 [8] | 22.2% |
| Total | 11 [1] | 9 [1] | 17 [2] | 24 [7] | 61 [11] | |
| % Vacuity | 9.1% | 11.1% | 11.8% | 29.2% | | |

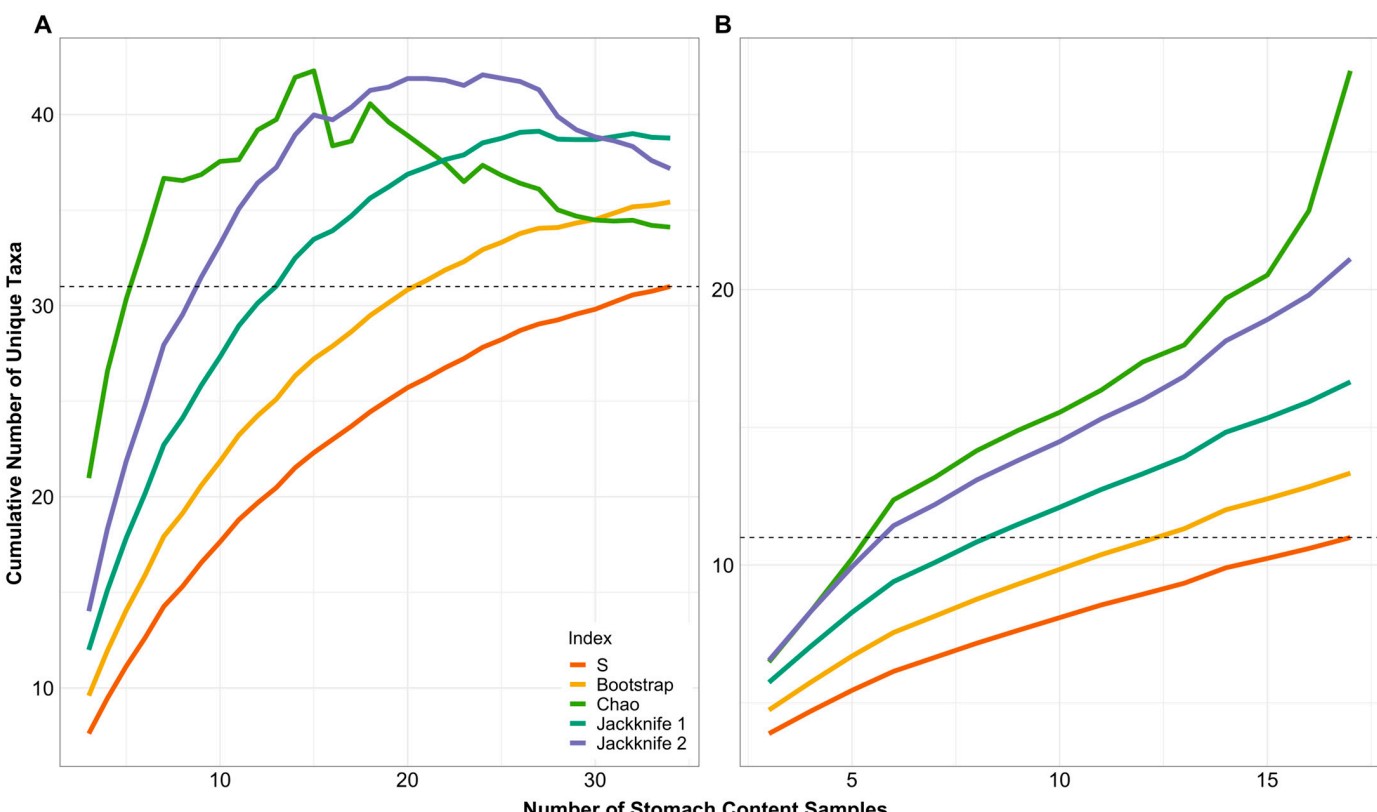

**Figure 4.** Cumulative prey curves using the cumulative number of prey items observed within the gut contents using modeled data from (**A**) the Atlantic Coast and (**B**) Gulf coast samples. The dotted line represents the total number of unique prey items observed within the gut contents. Colored symbology refers to the extrapolated species richness via specified model fits derived from the *poolaccum* function in the *vegan* package.

Of the 211 gastric lavage samples collected, 167 were deemed suitable for sequencing. Of those suitable samples, only 89 yielded high (>95%) sequence matches to vouchers available in genetic databases. The remaining 122 samples with low (<95%) sequence matches were identified visually by comparing to both photos of samples with high sequence reads and the invertebrate tissue guide. There were three classes of marine invertebrates observed within the gut contents: Bivalvia, Gastropoda and Malacostraca. Bivalves were the most frequently observed prey class, followed by gastropods and crustaceans (Table 3). Though observed in all locations, crustaceans had the greatest %FO in Fort Pierce, with the family Diogenidae being the most common crustacean observed. There were five families that were observed in all four sampling locations, including Cardiidae, Donacidae, Pinnidae, Busyconidae and Diogenidae. Cardiid bivalves had the greatest %FO of bivalve families in all locations except Fort Pierce, where lucinid clams had the greatest %FO. Busycon whelks had the greatest %FO for all Atlantic coast locations, whereas tulip snails from the family Fasciolariidae had the greatest %FO for gastropods on the Gulf coast. While Veneridae were observed in the gut contents of certain individuals, we had no positive identifications of hard clams (*Mercenaria* spp.). Other venerid species included cross barred venus (*Chione cancellata*), sunray venus clam (*Macrocallista nimbosa*) and lightning pitar (*Pitar fulminatus*). For immature rays, bivalves accounted for the greatest %FO (90.9%), followed by gastropods (50%) and crustaceans (4.5%), whereas in mature rays, gastropods and bivalves still accounted for the greatest %FO (79.3%), followed by an increased %FO for crustaceans (55.2%; Table 3; Figure 5). Cardiid bivalves had the greatest %FO for either life stage considered, whereas for gastropods, tulip snails had the greatest %FO for immature

rays and Busycon whelks for mature rays. Unidentified tissue accounted for a %FO of 18.2% and 17.2% for immature and mature rays, respectively.

**Table 3.** The % frequency of occurrence (%FO) of each prey family and class from rays caught in each location and life stage.

| Class | Family | Location | | | | Life Stage | |
|---|---|---|---|---|---|---|---|
| | | Sebastian | Fort Pierce | St. Lucie | Sarasota | Immature | Mature |
| Bivalvia | | 70.0 | 100 | 87.5 | 82.4 | 90.9 | 79.3 |
| | Arcidae | 30.0 | 0.0 | 0.0 | 0.0 | 9.1 | 3.4 |
| | Cardiidae | 40.0 | 12.5 | 62.5 | 64.7 | 63.6 | 41.4 |
| | Donacidae | 10.0 | 25.0 | 43.8 | 5.9 | 31.8 | 13.8 |
| | Lucinidae | 20.0 | 62.5 | 0.0 | 0.0 | 4.5 | 20.7 |
| | Noetiidae | 30.0 | 12.5 | 0.0 | 0.0 | 9.1 | 6.9 |
| | Pinnidae | 10.0 | 12.5 | 6.2 | 5.9 | 4.5 | 10.3 |
| | Semelidae | 10.0 | 0.0 | 12.5 | 0.0 | 9.1 | 3.4 |
| | Veneridae | 20.0 | 0.0 | 6.2 | 29.4 | 18.2 | 13.8 |
| Gastropoda | | 70.0 | 100 | 75 | 41.2 | 50.0 | 79.3 |
| | Aplysiidae | 0.0 | 12.5 | 0.0 | 0.0 | 4.5 | 0.0 |
| | Busyconidae | 40.0 | 50.0 | 31.2 | 5.9 | 13.6 | 37.9 |
| | Cerithiidae | 0.0 | 37.5 | 6.2 | 0.0 | 9.1 | 6.9 |
| | Fasciolariidae | 10.0 | 12.5 | 0.0 | 29.4 | 22.7 | 6.9 |
| | Melongenidae | 40.0 | 12.5 | 6.2 | 0.0 | 4.5 | 17.2 |
| | Muricidae | 0.0 | 12.5 | 0.0 | 0.0 | 4.5 | 0.0 |
| | Naticidae | 0.0 | 12.5 | 25.0 | 11.8 | 9.1 | 17.2 |
| | Olividae | 0.0 | 12.5 | 6.2 | 0.0 | 0.0 | 6.9 |
| | Strombidae | 0.0 | 12.5 | 18.8 | 0.0 | 0.0 | 13.8 |
| | Tonnidae | 0.0 | 0.0 | 6.2 | 0.0 | 0.0 | 3.4 |
| | Turbinidae | 0.0 | 0.0 | 12.5 | 0.0 | 4.5 | 3.4 |
| Malacostraca | | 40.0 | 62.5 | 31.2 | 17.6 | 4.5 | 55.2 |
| | Diogenidae | 30.0 | 50.0 | 31.2 | 17.6 | 4.5 | 48.3 |
| | Paguridae | 10.0 | 12.5 | 0.0 | 0.0 | 0.0 | 6.9 |
| Unknown | | 20.0 | 12.5 | 18.8 | 0.0 | 18.2 | 17.2 |

Using the accumulated weights, bivalves and gastropods were observed across all size ranges (Figure 5), whereas the crustacean proportion of the diet increased with size. The size classes by coast were unbalanced, where 79.3% of all mature rays (n = 28) were sampled in the Atlantic, and 52.6% of immature rays (n = 22) were sampled in the Atlantic. Despite having the highest %FO in Fort Pierce, crustaceans made up the largest proportion of diet by weight in Sebastian, when comparing all locations. Bivalves accounted for >50% of the total diet proportion only in Sarasota and St. Lucie, whereas bivalves and gastropods had similar proportions in both Fort Pierce and Sebastian. In all cases, the weight of the aggregated gut contents was less than 0.5% of the whitespotted eagle ray total body weight, following a negative trend with increasing size (Figure 6).

A few species accounted for the greatest %FO, %W and %IOI, including Atlantic giant cockle (*Dinocardium robustum*), pear whelk (*Fulguropsis pyruloides*), giant false donax (*Iphigenia brasiliana*) and crown conch (*Melongena corona*) (Table 4), where the %IOI was greater than 12%. The yellow prickly cockle (*Dallocardia muricata*) had a high %FO appearing in the gut contents of 22% of all individuals; however, despite a relatively low %W, there was a moderate %IOI of 5.14%. Two species with a moderate yet notable %FO, %W and %IOI include the thick lucine (*Phacoides pectinatus*) and banded tulip (*Cinctura hunteria*). Additionally, *Diogenidae* spp. had a large %FO (20%) and a moderate %IOI (6.69), likely due to contributions from both the thinstripe hermit crab (*Clibanarius vittatus*) and the giant red hermit crab (*Petrochirus diogenes*) that belong to Diogenidae. Both the sunray venus clam (*Macrocallista nimbosa*) and the lightning whelk (*Busycon sinistrum*) only appeared in a few sets of gut contents (<5), although their %W and %IOI were much greater than other species considered with a similar %FO. There was a relatively high %FO for unidentifiable

tissue, yet it accounted for a small %W and %IOI. Finally, the lowest %FO was observed for species in the families Aplysiidae, Muricidae, Naticidae and Tonnidae.

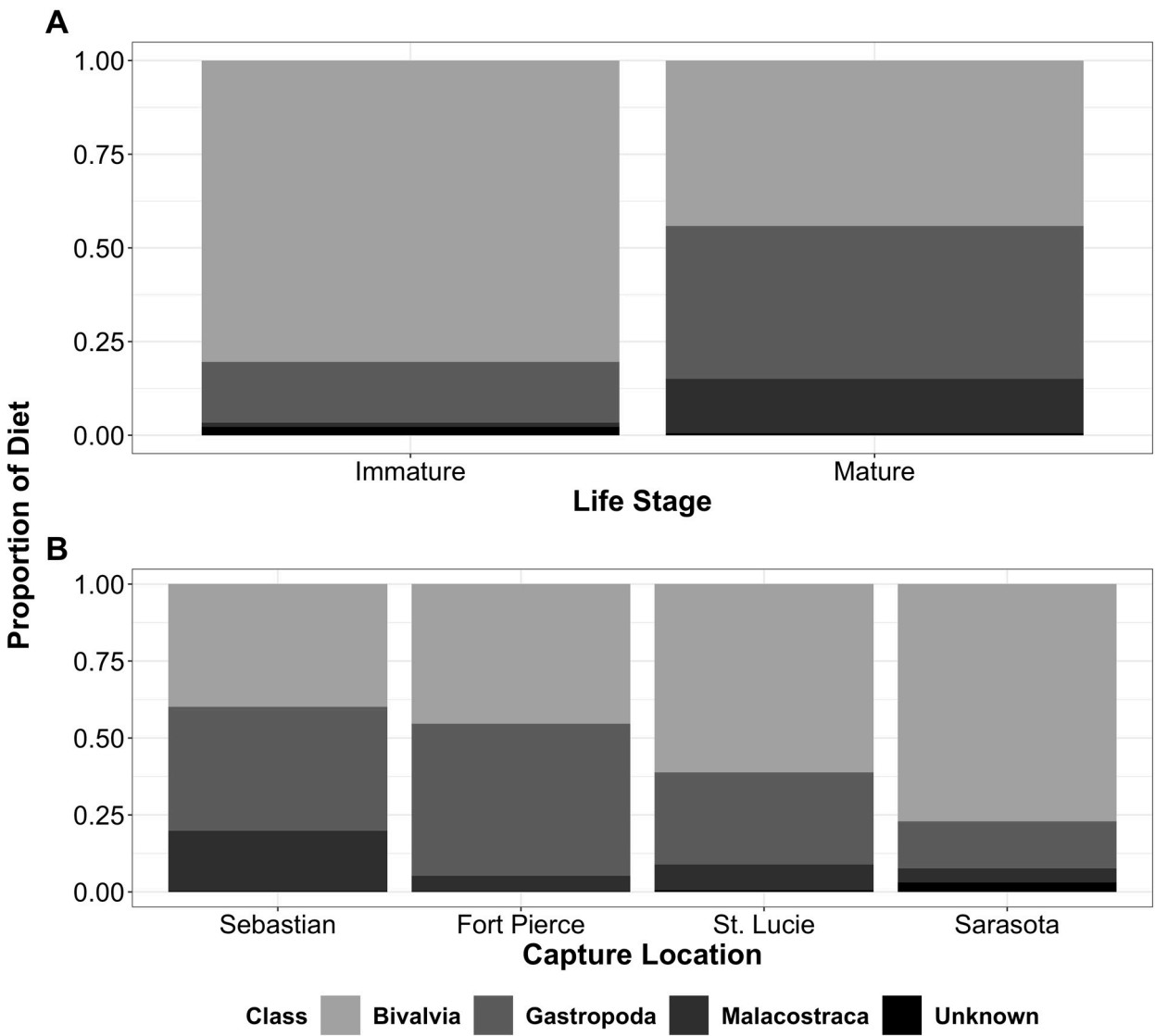

**Figure 5.** Proportion of diet by %W for each prey class observed in the gut contents represented by (**A**) life stage and (**B**) capture location.

Using the dietary data from all individuals, the PERMANOVA revealed a significant site effect (Pseudo-F = 3.007; $p$ = 0.001), but not a significant effect for life stage (Pseudo-F = 1.281; $p$ = 0.099) or the combined effects of life stage and location (Pseudo-F = 1.116; $p$ = 0.196). There was a high degree of overlap in the diet on the Atlantic coast (Figure 7). However, pairwise comparisons of location indicated that the diets of animals collected from St. Lucie were significantly different from that of both Fort Pierce (Pseudo-F = 2.040; $p$ = 0.036) and Sebastian (Pseudo-F = 2.463; $p$ = 0.006), but Fort Pierce and Sebastian were not significantly different from one another (Pseudo-F = 1.279; $p$ = 0.163). SIMPER analyses suggested one species, giant false donax (*I. brasiliana*), drove the most dissimilarity when comparing significant pairwise combinations of Atlantic locations. Contributions of this species to the diet were greater in St. Lucie than both Fort Pierce ($p$ = 0.020) and Sebastian ($p$ = 0.031; Table 5) locations. When comparing ray diets between St. Lucie and Fort Pierce, only one other species accounted for an average dissimilarity of >10%, which was the thick lucine (*P. pectinatus*), whereby the average abundance was significantly greater in Fort Pierce than St. Lucie ($p$ = 0.031). The yellow prickly cockle (*D. muricata*; $p$ = 0.035) was significantly

more abundant in St. Lucie than Fort Pierce, though the average abundance was <10%. All other species that were significantly dissimilar between Sebastian and St. Lucie were significantly more abundant in Sebastian, including cross-barred venus (*C. cancellata*, *p* = 0.017), lightning pitar (*P. fulminatus*, *p* = 0.035), ponderous ark (*Noetia ponderosa*; *p* = 0.008) and blood ark clam (*A. ovalis*; *p* = 0.003).

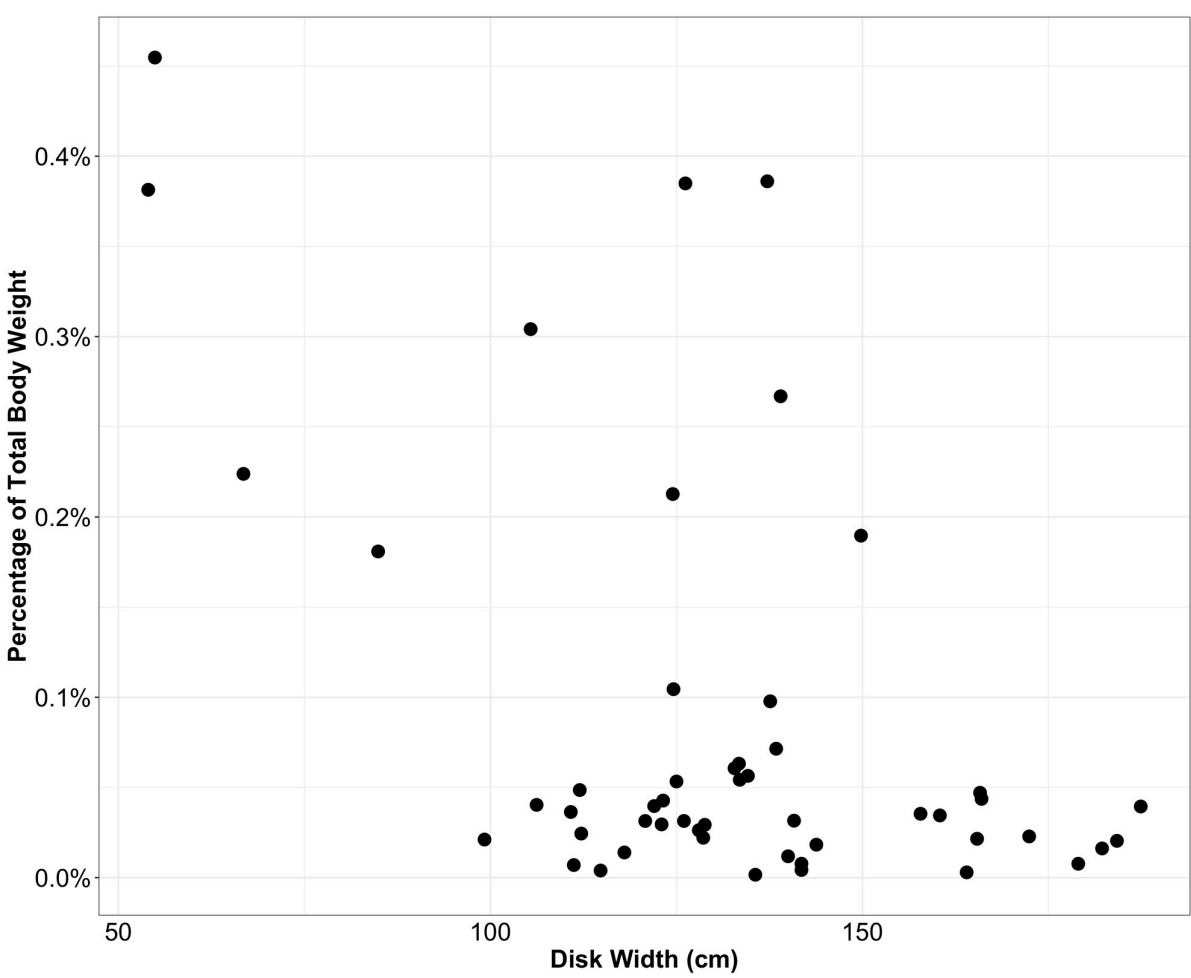

**Figure 6.** Percentage of total body weight (using the total gut content weight divided by the weight of each whitespotted eagle ray) with respect to whitespotted eagle ray disc width in cm.

**Table 4.** Prey identified in the gut contents down to lowest possible taxon. Index of importance (%IOI) calculated using frequency of occurrence (%FO) and percentage weight (%W).

| Class | Family | Prey Species | %FO | %W | %IOI |
|---|---|---|---|---|---|
| Bivalvia | Arcidae | *Anadara ovalis* | 6.00 | 0.18 | 0.11 |
| | Cardiidae | *Dinocardium robustum* | 28.00 | 24.56 | 68.76 |
| | | *Dallocardia muricata* | 22.00 | 2.34 | 5.14 |
| | Donacidae | *Iphigenia brasiliana* | 22.00 | 8.59 | 18.90 |
| | Lucinidae | *Phacoides pectinatus* | 14.00 | 4.89 | 6.85 |
| | Noetiidae | *Noetia ponderosa* | 8.00 | 0.60 | 0.48 |
| | Pinnidae | *Atrina rigida* | 8.00 | 0.27 | 0.22 |
| | Semelidae | *Semele purpurascens* | 6.00 | 0.67 | 0.40 |
| | Veneridae | *Macrocallista nimbosa* | 6.00 | 6.97 | 4.18 |
| | | *Chione cancellata* | 4.00 | 0.67 | 0.27 |
| | | *Pitar fulminatus* | 4.00 | 0.65 | 0.26 |
| | | *Veneridae* spp. | 4.00 | 0.44 | 0.18 |
| | Unidentified | *Unidentified bivalve* | 4.00 | 0.25 | 0.10 |

**Table 4.** *Cont.*

| Class | Family | Prey Species | %FO | %W | %IOI |
|---|---|---|---|---|---|
| Gastropoda | Aplysiidae | *Aplysia fasciata* | 2.00 | 0.04 | 0.01 |
| | Busyconidae | *Fulguropsis pyruloides* | 24.00 | 16.04 | 38.50 |
| | | *Busycon sinistrum* | 8.00 | 3.55 | 2.84 |
| | Cerithiidae | *Cerithium atratum* | 8.00 | 0.18 | 0.14 |
| | Fasciolariidae | *Cinctura hunteria* | 14.00 | 3.16 | 4.42 |
| | Melongenidae | *Melongena corona* | 12.00 | 10.23 | 12.28 |
| | Muricidae | *Stramonita canaliculata* | 2.00 | 0.02 | 0.00 |
| | Naticidae | *Neverita duplicata* | 6.00 | 0.83 | 0.50 |
| | | *Naticidae* spp. | 2.00 | 0.33 | 0.07 |
| | | *Polinices uber* | 2.00 | 0.24 | 0.05 |
| | | *Polinices* spp. | 2.00 | 0.13 | 0.03 |
| | | *Polinices lacteus* | 2.00 | 0.04 | 0.01 |
| | Olividae | *Oliva sayana* | 4.00 | 2.36 | 0.94 |
| | Strombidae | *Strombus alatus* | 6.00 | 1.23 | 0.74 |
| | Tonnidae | *Tonna galea* | 2.00 | 0.27 | 0.05 |
| | Turbinidae | *Turbo castanea* | 4.00 | 0.02 | 0.01 |
| | Unidentified | *Unidentified gastropod* | 2.00 | 0.24 | 0.05 |
| Malacostraca | Diogenidae | *Diogenidae* spp. | 20.00 | 3.34 | 6.69 |
| | | *Clibanarius vittatus* | 6.00 | 3.38 | 2.03 |
| | | *Petrochirus diogenes* | 4.00 | 2.30 | 0.92 |
| | Paguridae | *Pagurus pollicaris* | 4.00 | 0.45 | 0.18 |
| Unidentified | Unidentified | *Unidentified* | 16.00 | 0.55 | 0.88 |

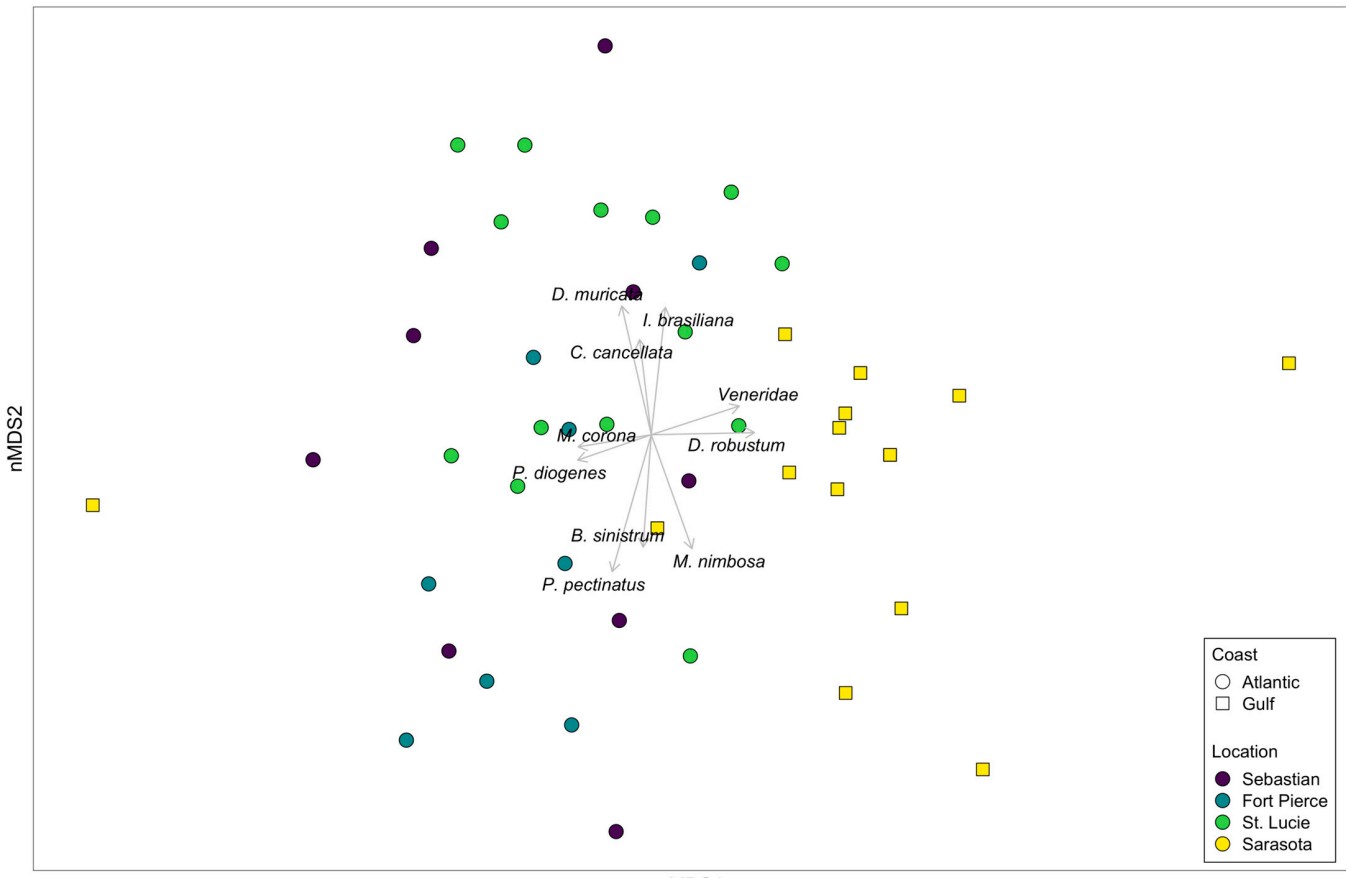

**Figure 7.** Non-metric multidimensional scaling (nMDS) plots showing dissimilarity among all individuals. Each point refers to an individual; colored symbology refers to each location and shape symbology refers to the coastline.

**Table 5.** Similarity percentage (SIMPER) results from prey abundances and their dissimilarity between two significant pairwise combination of locations. Average abundance (Av. Abund.) is described for each location, along with the average dissimilarity (Av. Diss.) and respective standard deviation (Diss/SD) between pairwise combinations, and the cumulative contribution (Cum.%) and *p*-values. Average abundance values were multiplied by 1.0 $E^3$ and dissimilarity values and cumulative contribution multiplied by 1.0 $E^2$.

| (A) Average dissimilarity Taxon | Sarasota Av. Abund. | Fort Pierce Av. Abund. | Av. Diss. | Diss/SD | Cum.% | *p* |
|---|---|---|---|---|---|---|
| *D. robustum* | 79.2 | 0.0 | 31.4 | 32.1 | 31.6 | 0.021 |
| *P. pectinatus* | 0.0 | 8.6 | 10.9 | 18.8 | 54.5 | 0.017 |
| (B) Average dissimilarity Taxon | Sarasota Av. Abund. | Sebastian Av. Abund. | Av. Diss. | Diss/SD | Cum.% | *p* |
| *D. robustum* | 79.2 | 5.8 | 30.0 | 30.0 | 30.5 | 0.023 |
| *C. cancellata* | 0.0 | 7.9 | 5.5 | 15.0 | 61.3 | 0.023 |
| *N. ponderosa* | 0.0 | 2.0 | 1.9 | 4.8 | 91.7 | 0.020 |
| *A. ovalis* | 0.0 | 2.3 | 1.1 | 2.2 | 96.3 | 0.006 |
| (C) Average dissimilarity Taxon | Sarasota Av. Abund. | St. Lucie Av. Abund. | Av. Diss. | Diss/SD | Cum.% | *p* |
| *D. robustum* | 79.2 | 3.0 | 30.7 | 29.8 | 31.4 | 0.001 |
| *I. brasiliana* | 0.1 | 18.5 | 17.3 | 25.0 | 49.2 | 0.022 |
| *D. muricata* | 0.0 | 5.2 | 7.5 | 14.2 | 75.8 | 0.011 |
| *C. hunteria* | 18.2 | 0.0 | 4.7 | 8.0 | 80.5 | 0.017 |
| (D) Average dissimilarity Taxon | Fort Pierce Av. Abund. | St. Lucie Av. Abund. | Av. Diss. | Diss/SD | Cum.% | *p* |
| *I. brasiliana* | 4.4 | 18.5 | 20.4 | 24.5 | 22.1 | 0.020 |
| *P. pectinatus* | 8.8 | 0.0 | 11.2 | 16.7 | 50.9 | 0.031 |
| *D. muricata* | 1.0 | 5.2 | 8.5 | 14.0 | 70.1 | 0.035 |
| (E) Average dissimilarity Taxon | Sebastian Av. Abund. | St. Lucie Av. Abund. | Av. Diss. | Diss/SD | Cum.% | *p* |
| *I. brasiliana* | 1.0 | 18.5 | 18.3 | 22.9 | 19.1 | 0.031 |
| *C. cancellata* | 7.9 | 0.0 | 6.0 | 14.0 | 63.0 | 0.017 |
| *P. fulminatus* | 14.9 | 0.3 | 6.0 | 16.7 | 69.3 | 0.035 |
| *N. ponderosa* | 2.0 | 0.0 | 2.0 | 4.3 | 89.6 | 0.008 |
| *A. ovalis* | 2.3 | 0.0 | 1.3 | 2.2 | 93.9 | 0.003 |

PERMANOVA pairwise comparisons of location indicated that the Gulf coast location (Sarasota) was significantly different from that of all Atlantic locations, including Sebastian (Pseudo-F = 2.798; *p* = 0.001), Fort Pierce (Pseudo-F = 3.334; *p* = 0.001) and St. Lucie (Pseudo-F = 4.959; *p* = 0.001). When comparing pairwise combinations of the Gulf to Atlantic locations, the Atlantic giant cockle (*D. robustum*) drove the most dissimilarity. *D. robustum* abundance was significantly greater in Sarasota than Fort Pierce (*p* = 0.021), Sebastian (*p* = 0.023) and St. Lucie (*p* = 0.001; Table 5). When comparing ray diets between Gulf and Atlantic locations, only two other species accounted for an average dissimilarity of >10%, including the thick lucine (*P. pectinatus*), whereby the average abundance was significantly greater in Fort Pierce than Sarasota (*p* = 0.017), and the giant false donax (*I. brasiliana*), which was significantly more abundant in St. Lucie than Sarasota (*p* = 0.022). The cross-barred venus (*C. cancellata*; *p* = 0.023), ponderous ark (*N. ponderosa*; *p* = 0.020) and blood ark clam (*A. ovalis*; *p* = 0.06) were all significantly more abundant in Sebastian than Sarasota. Finally, the yellow prickly cockle (*D. muricata*; *p* = 0.011) was significantly more abundant in St. Lucie than Sarasota, whereas the banded tulip (*C. hunteria*; *p* = 0.017) was significantly more abundant in Sarasota than St. Lucie.

## 4. Discussion

Our study is the first to quantitatively describe the whitespotted eagle ray diet in US waters. With the incorporation of DNA barcoding techniques to aid visual identification, our results suggest that whitespotted eagle rays are invertivores, with diets that vary by region across the state of Florida. Though mollusks have collectively comprised the majority of the species' diet in several previous studies in the northwest Atlantic [7,8,10–13], our work revealed, via non-lethal methods, species-level identification of these otherwise unrecognizable prey, and a significantly broader diet than has been previously described

at most locales of the region. Furthermore, we were able to assess and quantify the relative contributions of various invertebrate taxa to the diet of this poorly studied ray species, which provided opportunities to examine finer-scale feeding patterns in both space and ontogeny.

In total, we observed 33 unique prey items in the gut contents; however, our cumulative prey curve data from both coasts did not reach an asymptote, suggesting that the sample size was insufficient to fully describe the diet of whitespotted eagle rays. As such, we recommend future studies target larger sample sizes for analyses of this type and scale. Additionally, we caution that inter-study comparisons consider the methods of gut content acquisition. For example, work by Serrano-Flores et al. [13], who obtained gastrointestinal tracts from whole individuals landed in fisheries, indicated a much higher dominance of gastropods in the *A. narinari* diet, in particular *S. pugilis*, in the diet of rays collected off Campeche, MX. While we did encounter strombid conchs in the lavage contents here (*L. raninus* and *S. alatus*) and in Bermuda (*Strombus costatus* [12]), the tissues of these animals are considerably larger and tougher than those of the bivalves consumed, and thus may not have been fully dislodged from the gastrointestinal tract of the rays during the lavage process. Unlike Serrano-Flores et al. [13], we did not acquire opercula regularly, which they used to facilitate gastropod presence and identification in the gut contents. As such, we suspect tissues from larger gastropods could be underrepresented in this data set, and the relative importance of bivalves could be overestimated. We therefore suggest future studies sacrifice a portion of their animals following gastric lavage to better assess how this probability of prey identification can be influenced by extraction technique (i.e., lavage vs. dissection). However, non-lethal techniques such as sonography could also be used to assess the gastrointestinal tract for hard parts following pulsed gastric lavage.

Even among the four locations in Florida, with three locations on the Atlantic coast, we observed local variation in diet. This is not particularly surprising as previous studies of related taxa have indicated variability over estuarine scales at 10s of km [35], likely driven by local prey availability. Additionally, despite their capacity to undertake larger migrations, whitespotted eagle rays from the Atlantic coast exhibit strong affinities to the three coastal inlets examined in this study [36]. These affinities to the inlet habitats may suggest congruence between the dietary composition and local prey availability, though invertebrate abundance in each location has not been recently described. Prey availability could be attributed to differences in benthic habitat near the coastal inlets. Rays caught near the inlet in Sarasota and St. Lucie occurred on sandbars with minimal vegetation, in Fort Pierce on a mix of sand and mud bottom with sparse vegetation, and in Sebastian on a sand and mud bottom near oyster shoals. However, for all locations, the further from the inlet, the more the benthic structure consisted of mud and sparse vegetation.

Although they accounted for a smaller component of the overall diet, the presence of crustaceans (i.e., hermit crabs) is consistent with findings from rays caught in Mexico [13] and congeners (*A. ocellatus*) caught in the Indo-Pacific [14]. Additionally, crustacean %W increased with life stage similar to that observed in rays caught in Mexico [13], but not in the Indo-Pacific where consumption decreased with size. Here, that increase in hermit crab (and gastropod) consumption may correspond to the increase in dental plate and gape size. However, it is not known whether the consumption of hermit crabs by whitespotted eagle rays is intentional or incidental. Hermit crabs collected locally were found to occupy the shells of gastropods (e.g., crown conch, lightning whelk, pear whelk and banded tulip), the tissues of which were all identified in the gut contents of the rays sampled. In the Indian River Lagoon, the thinstriped hermit crab (*C. vittatus*) preferentially and intentionally kills crown conchs (*M. corona*) for their shells when they are available [37]. Hermit crab occurrence was highest in ray diets at the Sebastian and Fort Pierce study sites, where abundances of *C. vittatus* typically peak during summer [38,39]. However, these locations are also where we observed the highest frequency of occurrence of crown conchs in the diet, suggesting the rays may be targeting these gastropods but are instead consuming the hermit crab occupants of their shells. More research is needed to understand

this complex, multi-species, ecological interaction. Indeed, hermit crabs have not only been reported in the diet of *Aetobatus* spp., but they are also found in stomach contents of bullnose rays (*Myliobatis freminvillei*), from Delaware Bay, which are also known to prey on gastropods [40]. These authors also speculated over the incidental consumption of hermit crabs (*Pagurus* spp.) but found their elevated importance values to be too high to represent coincidental interactions [40]. It is not known whether rays can distinguish between shells occupied by conchs and those occupied by hermit crabs with their electrosensory system, although this presents an interesting line of future research.

Though the effect of life stage on dietary composition was not considered significant, and could be a result of insufficient sample sizes, young whitespotted eagle rays had higher dietary proportions of bivalves than mature conspecifics. This observation is consistent with findings from other durophagous rays (e.g., cownose [35]), and may be explained by ontogenetic differences in distribution patterns and/or more fine-scale feeding habitat preferences [41]. For example, DeGroot et al. [36] showed that younger individuals spend more time occupying inshore regions of the Indian River Lagoon, while adults frequent inlets and nearshore areas. These distributional differences likely correlate with sediment composition as well, with younger rays feeding in the finer sediment that typifies inshore lagoonal areas, while adults likely feed in coarser sand habitats that characterize inlet and offshore areas and are likely more challenging to excavate. These habitat differences also likely drive differences in dominant prey taxa, with softer sediment and more freshwater influenced estuaries supporting higher densities of bivalves, while more saline conditions facilitate higher densities of gastropods. Some of these gastropods, which also have thicker shells, may only be capable of being consumed by adults since bite force is known to scale with ontogeny in similar species [42]. In conclusion, the ontogenetic differences in diet in *A. narinari* may be driven by a combination of both ecological and morphological constraints between the two life stages.

Benthopelagic batoids have long been considered threats to shellfish populations [43–46], to the extent that cownose rays have been culled to reduce their populations with the intention of reducing their impact on shellfisheries. However, modeled trophic relationships including shellfisheries, cownose ray populations and large coastal shark populations do not suggest that a reduction in ray populations corresponds to improved shellfisheries [47]. Additionally, previous studies of cownose rays employed DNA barcoding to assess the presence of commercially important bivalve species in the diet [16]. They compared the stomach and gut contents of cownose rays to known bivalves in the mid-Atlantic region, using genomic markers to resolve if the rays were foraging on a variety of economically important species (i.e., hard clams (*M. mercenaria*), bay scallops (*Argopecten irradians*), eastern oysters (*Crassostrea virginica*), etc.). While seven different species were considered, only two species were found within the digestive tract, including the commercially important soft-shell clam (*Mya arenaria*) and stout tagelus (*Talegus plebius*). Similarly, even in regions with shellfish enhancements (i.e., Sebastian with hard clam and eastern oyster aquaculture operations, Sarasota with hard clam restoration), we did not observe any high (>95%) sequence matches with any commercially important bivalves (publicly available in GenBank) among any of our locations; however, we did observe arks (*A. ovalis* and *N. ponderosa*) and sunray venus (*M. nimbosa*) clams in the diet. There is growing interest in the culture of these bivalve species in Florida [48–50], and there are growout operations emerging throughout the state. Two congeneric species, the calico clam (*Macrocallista maculata*) and eared ark clam (*Anadara notabilis*), have also been shown to comprise the diet of rays from Bermuda [12], suggesting the importance of these genera to the diet of whitespotted eagle rays across larger scales. Thus, interactions between whitespotted eagle rays and these less popular species in shellfish culture remains a possibility, as is the consumption of commercially and recreationally important bivalves such as bay scallops and oysters in areas where these shellfish are naturally more abundant. This includes regions like Florida's big bend, where whitespotted eagle rays occur seasonally [36]. However, it is worth noting that the diet analysis also revealed the high importance of predatory gastropods (*M. corona* and

*F. pyruloides*) that have been shown to consume some of these bivalves [51,52], highlighting the variable role *A. narinari* plays in structuring benthic mollusk communities through direct and indirect consumptive effects.

Despite various testing efforts of different protocols (e.g., different DNA extraction methods/kits, taq polymerase, primers), only 56.3% of our samples yielded high (>95%) sequence matches. This could be a result of a variety of different factors, including freeze/thaw cycles allowing the degradation of tissue, or it could have been a result of the high mucopolysaccharide content of mollusks [53] preventing successful extraction or amplification. The addition of a potassium acetate step during DNA extraction is typically used to aid in cell lysis, and the precipitation of proteins and polysaccharides [54]. It is possible that adding potassium acetate in addition to an overnight proteinase K digestion and mechanical homogenization was not enough to break down the mucopolysaccharides. However, almost all samples could be identified to a class level when comparing sample photos to those of the high (>95%) sequence identifications in conjunction with the invertebrate tissue guide. Regardless, the few samples that could not be sequenced or visually identified, could potentially match species that have not yet been sequenced or documented in genetic databases. Future studies considering this technique should place greater emphasis on barcoding all prospective prey items, though most species encountered during opportunistic invertebrate collections had already been sequenced and were available.

We employed DNA barcoding techniques to account for gut content weight by prey type; however, one limitation of our methodology could be potential misidentification when initially sorting gut contents. One strategy in diet analyses to avoid the misidentification of visually identified prey items is to employ metabarcoding techniques, whereby gut contents would be homogenized and processed via bulk DNA extraction [55], thereby removing the bias of processing gut contents visually and reducing the effort required for each sample. Similarly, this methodology has been successfully conducted using less invasive methods to assess the elasmobranch diet using cloacal swabs [56]. However, metabarcoding can also unintentionally remove low-abundance sequences from the dataset [57] and has challenges with translating reads to quantities, which are important for determining the dietary importance of certain prey taxa.

Alternatively, while DNA analyses provide a refined insight into dietary information, stable isotope analysis is a chemical ecological approach to track long-term, assimilated dietary habitats and energy flow within food webs through analyses of nitrogen, carbon and sulfur ($\delta^{15}$N, $\delta^{13}$C and $\delta^{34}$S) isotopes. As such, analyses of key stable isotopes from whitespotted eagle ray tissues and invertebrate prey items can quantitatively characterize trophic position and track basal carbon sources of prey [58]. Such data also provide opportunities to examine how trophic position and carbon sources vary across ray ontogeny (YOY to mature) and space, in the absence of gut contents [59], though it is limited in its ability to identify important prey sources. Moreover, understanding whitespotted eagle ray reliance on specific prey items can offer valuable insights into the toxin pathways in these ecosystems in the face of harmful algal blooms [60]. For example, blooms of *Karenia brevis* are becoming more common off Sarasota, and toxins (i.e., brevetoxins) from this dinoflagellate have been previously shown to transfer in both bivalve and gastropod prey identified here [61,62], suggesting that whitespotted eagle rays are likely exposed as well. As such, further research should assess long-term assimilated dietary patterns to help decipher common toxin pathways, allowing us to understand how these animals can be affected in the face of increasing threats to marine ecosystems.

## 5. Conclusions

Here, we used DNA barcoding in conjunction with the visual identification of gut contents using the non-lethal method of pulsed gastric lavage. These data suggested some dietary overlap in common prey families between the Atlantic and Gulf coasts of Florida; however, we also observed large variation in the most frequently identified families among locations on the Atlantic coast, suggesting that while whitespotted eagle rays can



be considered invertivores, their diet varies substantially by location. Finally, though durophagous rays are largely thought to interact with shellfish enhancement activities in Florida, no common commercially important bivalves were observed in the gut contents of whitespotted eagle rays, although gastropod predators of these bivalves were detected in the diet, suggesting rays can play a facilitative role at reducing predation threats on shellfish aquaculture operations.

**Supplementary Materials:** The following supporting information can be downloaded at: https://zenodo.org/record/8092612 (accessed on 27 June 2023), Document S1: Gut content visual identification catalog from all individuals considered.

**Author Contributions:** B.V.C., K.B.-H. and M.J.A. conceived the study. B.V.C., K.B.-H. and M.J.A. obtained funding. R.J.E. and J.D.V. provided reagents and equipment for DNA analysis. B.V.C., K.B.-H., T.J.O., B.C.D. and M.J.A. conducted fieldwork. B.V.C. conducted lab work with input from R.J.E., B.V.C. and T.J.O. processed sequence data and analyzed visual identification photo catalog. B.V.C. and M.J.A. conducted analysis. B.V.C. wrote the manuscript. All authors have read and agreed to the published version of the manuscript.

**Funding:** Funding for this study was provided by the Mote Scientific Foundation, Georgia Aquarium, and the Indian River Lagoon Graduate Research Fellowship administered by the Harbor Branch Oceanographic Institute Foundation.

**Institutional Review Board Statement:** All animal collection and sampling procedures were approved by Florida Atlantic University's Institutional Animal Care and Use Committees (Animal Use Protocol #A19-26) and Florida Fish and Wildlife Conservation Commission Special Activity License permit numbers (SAL-19-1785-SRP, SAL-20-1785-SRP, SAL-21-1785-SRP), as well as Mote Marine Laboratory's Institutional Animal Care and Use Committees (Animal Use Protocol 19-10-KBH1 and 20-09-KBH1) and Florida Fish and Wildlife Conservation Commission Special Activity License permit numbers (SAL-19-1140-SRP and SAL-20-1140-SRP). All efforts were made to minimize animal suffering during tagging procedures.

**Data Availability Statement:** All data presented in this study are available in a public repository on Github (https://github.com/briannacahill/SERDiet).

**Acknowledgments:** We would like to thank the various graduate students and field support staff who made our sampling efforts possible, particularly members of the Fisheries Ecology and Conservation Lab at FAU-HBOI (M. McCallister, A. Rojas-Corzo, K. McCulloch, S. Lombardo, L. Brewster, M. Edwards) and Mote Marine Laboratory (P. Hull, D. Dougherty, G. Byrd, C. Cole, K. Wilkinson, J. Wolfe, V. Hagan, J. Morris). Additionally, we thank I. Segura-Garcia, C. Santamaria, L. Hoopes and C. Perricone for their guidance and constructive feedback.

**Conflicts of Interest:** The authors declare no conflict of interest. The funders had no role in the design of the study; in the collection, analyses, or interpretation of data; in the writing of the manuscript; or in the decision to publish the results.

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
