# Peer review of "Diet and Feeding Ecology of the Whitespotted Eagle Ray (Aetobatus narinari) from Florida Coastal Waters Revealed via DNA Barcoding"

_fishes, doi:10.3390/fishes8080388_

Round 1
Reviewer 1 Report
The paper presents interesting information about the diet of a protected species in Florida Coastal waters. In addition to the new information, it also investigates the role of this species in the marine ecosystem and its potential impact on the restoration of commercially important populations of bivalves.
I have concerns about the data available for some of the analysis due to the poor number of samples. For example, a %FO of occurrence of 2.00 means that the species only occurred in one stomach; very low %FO are presented for most of the species. The authors assume this pitfall in the discussion but maybe more debate on this can be added.
I have other minor suggestions presented below.
Suggestions
1. Line 79. Remove “For example”
2. End of Introduction, objectives. Suggestion to refer here that non-lethal techniques were used.
3. Material and methods, sample size sufficiency curves. Change to “Species accumulation/cumulation curves”?
4. Results. Consider presenting disc width ranges by area instead of means.
5. Results, page 10, lines 26. Genetic databases often have misidentified species. Did the authors consider barcoding the individuals from the “invertebrate collection”?
6. Results, page 12, lines 30. Remove YOY from the sentence “unbalanced (…)” as you have very few individuals.
7. Results, page 12, lines 30. Include the number of individuals for immature and mature rays.
8. Discussion, page 17, lines 39. Here authors refer to species cumulative prey curves (see 3.). In addition, shouldn´t these curves be analysed by area (at least Atlantic vs. Sarasota)?
9. Discussion, page 17, lines 39. The low number of samples can lead to some bias in the results. For example, ontogenetic variation in diet is likely to occur (is indeed discussed) but analysis do not identify such variation.
10. Discussion/conclusion. Although I am not familiar with marine communities in Florida, I do not expect them to be similar in the Atlantic and in the Gulf of Mexico. Thus, it is not surprising that diet differs between the two main sampled areas. When concluding “their diet varies substantially by location” more information on marine communities from both areas should be added as authors are comparing areas with different conditions. In the 3rd paragraph of the discussion (lines 41, page 17), the authors attribute differences to “prey availability” but this can be better discussed.
Author Response
Reviewer 1:
Comments and Suggestions for Authors
The paper presents interesting information about the diet of a protected species in Florida Coastal waters. In addition to the new information, it also investigates the role of this species in the marine ecosystem and its potential impact on the restoration of commercially important populations of bivalves.
I have concerns about the data available for some of the analysis due to the poor number of samples. For example, a %FO of occurrence of 2.00 means that the species only occurred in one stomach; very low %FO are presented for most of the species. The authors assume this pitfall in the discussion but maybe more debate on this can be added.
RESPONSE: We have added more information about the low %FO in the results and discussion.
L332-333: “Additionally, the lowest %FO was observed for species in the families Aplysiidae, Muricidae, Naticidae and Tonnidae.”
I have other minor suggestions presented below.
Suggestions
- Line 79. Remove “For example”
RESPONSE: Accepted.
L82-85: “The cytochrome c oxidase subunit 1 gene (CO1) is a well-known genetic marker for animals [20], making it an ideal gene to utilize for DNA barcoding of whitespotted eagle ray stomach contents, which are generally devoid of accompanying mollusk shells.”
- End of Introduction, objectives. Suggestion to refer here that non-lethal techniques were used.
RESPONSE: Accepted.
L102-105: “Here, we used a combination of DNA barcoding techniques and visual identification to describe the diet of whitespotted eagle rays from Florida coastal waters using non-lethal collection techniques.”
- Material and methods, sample size sufficiency curves. Change to “Species accumulation/cumulation curves”?
RESPONSE: Accepted.
L211-213: “Species accumulation curves were generated using the vegan package [29] to assess the total number of unique prey items collected from stomach contents for all rays sampled [14,30,31].”
- Consider presenting disc width ranges by area instead of means.
RESPONSE: We thank the reviewer for this feedback. We added in the ranges but also kept the means.
L238-241: “Disc width of the sampled animals ranged from 66.8 – 182.2 cm (131.9 ± 9.2 cm; n = 10) in Sebastian, 125.0 – 187.4 cm (142.5 ± 8.1 cm; n = 8) in Fort Pierce, 99.2 – 166.0 cm (134.1 ± 3.4 cm; n = 16) in St. Lucie and 54.0 – 184.2 cm (123.7 ± 10.0 cm; n = 17) in Sarasota.”
- Results, page 10, lines 26. Genetic databases often have misidentified species. Did the authors consider barcoding the individuals from the “invertebrate collection”?
RESPONSE: We did try barcoding the individuals from our invertebrate collection efforts; however, we only preserved a small portion of the tissue after photographing for DNA barcoding. Unfortunately, these tissues were used early on when troubleshooting DNA extraction/PCR protocols and we didn’t end up collecting more, as most of the species we found had been sequenced and available in GenBank and BOLD.
L519-522: “Future studies considering this technique should place greater emphasis on barcoding all prospective prey items, though most species encountered during opportunistic invertebrate collections had already been sequenced and were available.”
- Results, page 12, lines 30. Remove YOY from the sentence “unbalanced (…)” as you have very few individuals.
RESPONSE: We thank the reviewer for pointing this out.
L306-308: “The size classes by coast were unbalanced, where 79.3% of all mature rays (n = 28) were sampled in the Atlantic, and 52.6% of immature rays (n = 22) were sampled in the Atlantic.”
- Results, page 12, lines 30. Include the number of individuals for immature and mature rays.
RESPONSE: We thank the reviewer for pointing this out.
L306-308: “The size classes by coast were unbalanced, where 79.3% of all mature rays (n = 28) were sampled in the Atlantic, and 52.6% of immature rays (n = 22) were sampled in the Atlantic.”
- Discussion, page 17, lines 39. Here authors refer to species cumulative prey curves (see 3.). In addition, shouldn´t these curves be analysed by area (at least Atlantic vs. Sarasota)?
RESPONSE: We have separated the cumulative prey curves by coast but the overall trend did not change.
L249-252: “Using these data, we fitted cumulative prey curves to the Atlantic and Gulf coast; however, after fitting multiple model structures to our data, the linear regression suggested insufficient sample size for the Atlantic (R2 = 0.991; p = 0.003; S = 31) and Gulf coast (R2 = 0.996; p = 0.001; S = 11).”
L268-272: “Cumulative prey curves using the cumulative number of prey items observed within the stomach contents using modeled data from the A) Atlantic Coast and B) Gulf coast samples. The dotted line represents the total number of unique prey items observed within the stomach contents. Colored symbology refers to the extrapolated species richness via specified model fits derived from the poolaccum function in the vegan package.”
- Discussion, page 17, lines 39. The low number of samples can lead to some bias in the results. For example, ontogenetic variation in diet is likely to occur (is indeed discussed) but analysis do not identify such variation.
RESPONSE: We have added additional clarification to address this point.
L456-458: “Though the effect of life stage on dietary composition was not considered significant, and could be a result of insufficient sample sizes, young whitespotted eagle rays had higher dietary proportions of bivalves than mature conspecifics.”
- Discussion/conclusion. Although I am not familiar with marine communities in Florida, I do not expect them to be similar in the Atlantic and in the Gulf of Mexico. Thus, it is not surprising that diet differs between the two main sampled areas. When concluding “their diet varies substantially by location” more information on marine communities from both areas should be added as authors are comparing areas with different conditions. In the 3rdparagraph of the discussion (lines 41, page 17), the authors attribute differences to “prey availability” but this can be better discussed.
RESPONSE: We have added additional information to address this point.
L425-430: “Prey availability could be attributed to differences in benthic habitat near the coastal inlets. Rays caught near the inlet in Sarasota and St. Lucie occurred on sandbars with minimal vegetation, in Fort Pierce a mix of sand and mud bottom with sparse vegetation, and in Sebastian on sand and mud bottom near oyster shoals. However, for all locations, the further from the inlet, the more the benthic structure consisted of mud and sparse vegetation.”
Reviewer 2 Report
The manuscript describes an interesting approach for examining the diet and feeding ecology of an endangered ray species, Aetobatus narinari, from off Florida. The data provides new insights into the diet of this highly mobile but mostly poorly managed batoid species. The interactions with shellfish populations are of particular interest considering that these rays are regularly considered a threat to these industries I still have a number of minor issues that need to be addressed before the manuscript can be accepted:
1) The Introduction starts with a flaw: the first sentence cites an incorrect reference as reference 1. does not include the content of the first sentence. Furthermore, the authors and the title of reference 1. are not correct and the publisher and place of publication are missing. The authors can find recent data for the numbers of batoid species, families and orders, e.g., in the following references:
-Last PR, White WT, de Carvalho MR, Séret B, Stehmann MFW, Naylor GJP (eds) (2016) Rays of the world. CSIRO Publishing, Melbourne.
-Weigmann S (2016) Annotated checklist of the living sharks, batoids and chimaeras (Chondrichthyes) of the world, with a focus on biogeographical diversity. J Fish Biol 88(3):837–1037
with updated numbers published in the following paper:
-Weigmann S (2017) Reply to Borsa (2017): comment on ‘annotated checklist of the living sharks, batoids and chimaeras (Chondrichthyes) of the world, with a focus on biogeographical diversity by Weigmann (2016)’. J Fish Biol 90(4):1176–1181.
2) The Introduction needs to be improved as basic biological data for Aetobatus narinari, such as maximum size, depth distribution, geographic distribution etc. are missing.
3) Species authorities should be added upon first mentioning of a species name throughout the manuscript. Depending on the journal, this might be done without indicating the year of the original description (usually the reference is not cited in the list of references then) or with indication of the year of the original description, with the reference to be included in or excluded from the list of references.
4) Reference [3] in line 42 should be moved to behind the end of the sentence as the Gulf of Mexico was actually the area investigated by those authors.
5) Line 43: Abbreviation IUCN needs to be explained.
6) Lines 127–129: Although sizes are indicated, it needs to be added if these refer to disc widths (what I assume).
7) Header of Table 1: You describe the units for DW and ray weight but not for stomach content weight. Please be consistent and either delete the units for all three (as the units are also indicated in the table itself) or give them for all three measurements.
8) Figure 4: The legend needs to be explained in the figure caption.
9) Lines 332 etc.: Not sure if I overlooked it but I did not see any indication of the level of significance (p value) used. This has to be included.
10) Caption to Figure 7: Abbreviation “det” (or “DET”?) needs to be explained and the authors might also want to introduce an abbreviation for “Non-metric multidimensional scaling plots”.
11) Lines 407–410: What about using other non-lethal methods such as ultrasound imaging (sonography) to detect such hard parts instead of sacrificing the animals?
12) The Conclusions section is rather long for a conclusions and should possibly be shortened to only include the most important conclusions, while others should be limited to the Discussion section.
Author Response
Reviewer 2:
Comments and Suggestions for Authors
The manuscript describes an interesting approach for examining the diet and feeding ecology of an endangered ray species, Aetobatus narinari, from off Florida. The data provides new insights into the diet of this highly mobile but mostly poorly managed batoid species. The interactions with shellfish populations are of particular interest considering that these rays are regularly considered a threat to these industries I still have a number of minor issues that need to be addressed before the manuscript can be accepted:
1) The Introduction starts with a flaw: the first sentence cites an incorrect reference as reference 1. does not include the content of the first sentence. Furthermore, the authors and the title of reference 1. are not correct and the publisher and place of publication are missing. The authors can find recent data for the numbers of batoid species, families and orders, e.g., in the following references:
-Last PR, White WT, de Carvalho MR, Séret B, Stehmann MFW, Naylor GJP (eds) (2016) Rays of the world. CSIRO Publishing, Melbourne.
-Weigmann S (2016) Annotated checklist of the living sharks, batoids and chimaeras (Chondrichthyes) of the world, with a focus on biogeographical diversity. J Fish Biol 88(3):837–1037
with updated numbers published in the following paper:
-Weigmann S (2017) Reply to Borsa (2017): comment on ‘annotated checklist of the living sharks, batoids and chimaeras (Chondrichthyes) of the world, with a focus on biogeographical diversity by Weigmann (2016)’. J Fish Biol 90(4):1176–1181.
RESPONSE: We thank the reviewer for catching this mistake and have since adjusted the reference.
L596-597: “Last PR, White WT, de Carvalho MR, Séret B, Stehmann MFW, Naylor GJP (eds). Rays of the world. Melbourne: CSIRO Publishing; 2016.”
2) The Introduction needs to be improved as basic biological data for Aetobatus narinari, such as maximum size, depth distribution, geographic distribution etc. are missing.
RESPONSE: We have added additional information on the maximum size and depth distribution. Information on geographic distribution was previously mentioned (see below).
L41-44: “For example, whitespotted eagle rays (Aetobatus narinari) are highly mobile, benthopelagic rays common in the western Atlantic, including the Gulf of Mexico [4] of which can dive to depths of 50.5 m but primarily occupy the upper 10 m of the water column [5] and reach disc widths of 2 m [6].”
3) Species authorities
RESPONSE: We opted to not include species authorities since it is not something required by the journal and is not required in the American Fisheries Society style guide on species names.
4) Reference [3] in line 42 should be moved to behind the end of the sentence as the Gulf of Mexico was actually the area investigated by those authors.
RESPONSE: We thank the reviewer for pointing this out.
L41-44: “For example, whitespotted eagle rays (Aetobatus narinari) are highly mobile, benthopelagic rays common in the western Atlantic, including the Gulf of Mexico [4] of which can dive to depths of 50.5 m but primarily occupy the upper 10 m of the water column [5] and reach disc widths of 2 m [6].”
5) Line 43: Abbreviation IUCN needs to be explained.
RESPONSE: Accepted.
L44-48: “The International Union for Conservation of Nature (IUCN) Red List of Threatened Species classifies whitespotted eagle rays as “Endangered” [7] with a decreasing trend in population, though both Florida and Alabama have implemented protections for whitespotted eagle rays, preventing targeted fisheries and harvest.”
6) Lines 127–129: Although sizes are indicated, it needs to be added if these refer to disc widths (what I assume).
RESPONSE: Accepted.
L130-133: “Rays designated as young of year (<70 cm disc width [24]), hereafter referred to as YOY, received the smallest diameter hose of 9.5 mm; immature rays (70 – 127 cm) received the medium diameter hose of 12.7 mm; mature rays (>127 cm [6]) received the largest diameter hose of 15.8 mm.”
7) Header of Table 1: You describe the units for DW and ray weight but not for stomach content weight. Please be consistent and either delete the units for all three (as the units are also indicated in the table itself) or give them for all three measurements.
RESPONSE: We thank the reviewer for catching this mistake.
L253-255: “Disc width (DW) is shown in cm, whitespotted eagle ray weight in kg and stomach content weight (SC Weight) in g refers to the total weight of all tissues present in the stomach contents.”
8) Figure 4: The legend needs to be explained in the figure caption.
RESPONSE: Accepted.
L271-272: “Colored symbology refers to the extrapolated species richness via specified model fits derived from the poolaccum function in the vegan package.”
9) Lines 332 etc.: Not sure if I overlooked it but I did not see any indication of the level of significance (p value) used. This has to be included.
RESPONSE: We thank the reviewer for catching this mistake.
L234-235: “The threshold for significance was defined as α = 0.05.”
10) Caption to Figure 7: Abbreviation “det” (or “DET”?) needs to be explained and the authors might also want to introduce an abbreviation for “Non-metric multidimensional scaling plots”.
RESPONSE: We thank the reviewer for catching this typo. Additionally, we have introduced an abbreviation for “Non-metric multidimensional scaling plots.”
L358-359: “Non-metric multidimensional scaling (nMDS) plots showing dissimilarity among all individuals.”
11) Lines 407–410: What about using other non-lethal methods such as ultrasound imaging (sonography) to detect such hard parts instead of sacrificing the animals?
RESPONSE: We had not considered sonography and thank the reviewer for mentioning it, though we did not consider as the expensive equipment required is not always available to those not studying reproductive biology.
L415-417: “However, non-lethal techniques such as sonography could also be used to assess the gastrointestinal tract for hard parts following pulsed gastric lavage.”
12) The Conclusions section is rather long for a conclusions and should possibly be shortened to only include the most important conclusions, while others should be limited to the Discussion section.
RESPONSE: We agree with the reviewer and have adjusted the conclusions section to only reflect the most important conclusions.
L554-564: “Here, we used DNA barcoding in conjunction with visual identification of stomach contents using the non-lethal method of pulsed gastric lavage. These data suggested some dietary overlap in common prey families between the Atlantic and Gulf coasts of Florida; however, we also observed large variation in the most frequently identified families among locations on the Atlantic coast, suggesting that while whitespotted eagle rays can be considered invertivores, their diet varies substantially by location. Finally, though durophagous rays are largely thought to interact with shellfish enhancement activities in Florida, no common commercially important bivalves were observed in the stomach contents of whitespotted eagle rays, although gastropod predators of these bivalves were detected in the diet, suggesting rays can play a facilitative role at reducing predation threats on shellfish aquaculture operations.”
Reviewer 3 Report
A nicely written piece of work. I have made some comments/suggestions marked on the manuscript (attached)
Authors may consider reconsidering the tables, figure sand text as some information is repeated in all three formats. What is the best way to present the data?

Author Response
Reviewer 3:
Comments and Suggestions for Authors
A nicely written piece of work. I have made some comments/suggestions marked on the manuscript (attached)
Authors may consider reconsidering the tables, figure sand text as some information is repeated in all three formats. What is the best way to present the data?
Suggestions
- End of page 1, “frequency”
RESPONSE: We are unsure what the reviewer was referring to with this comment.
- Line 238-239: with 3 YOY, how did you calculate 25% vacuity, rather than 33.3%?
RESPONSE: We appreciate this feedback (alongside comment #5) and amended table 2 to provide further clarification on the number of individuals without stomach contents. Rather than adding an additional row and column, the number of individuals without stomach contents is denoted in the square brackets within the total sections. We also added additional clarification to the results.
L246-247: “We only captured three YOY (<70 cm, n = 3) with stomach contents, thus all immature rays (<127 cm) were grouped for statistical analyses.”
L259-260: “Number of rays without stomach contents is shown within the square brackets ([ ]) for each location and life stage.”
- Line 242 (after “fit”): change to “fitted”
RESPONSE: Accepted.
L249-252: “Using these data, we fitted cumulative prey curves to the Atlantic and Gulf coast; however, after fitting multiple model structures to our data, the linear regression suggested insufficient sample size for the Atlantic (R2 = 0.991; p = 0.003; S = 31) and Gulf coast (R2 = 0.996; p = 0.001; S = 11).”
- Line 243: change “curve” to “curves”
RESPONSE: Accepted.
L249-252: “Using these data, we fitted cumulative prey curves to the Atlantic and Gulf coast; however, after fitting multiple model structures to our data, the linear regression suggested insufficient sample size for the Atlantic (R2 = 0.991; p = 0.003; S = 31) and Gulf coast (R2 = 0.996; p = 0.001; S = 11).”
- Table 2, line 250: Maybe add a column of the number of rays with zero stomach contents (empty) for clarity? I see the explanation table 2 caption. Still needs clarifying here as you did not only catch 3 YOY, but 4 (1 with no stomach contents)
RESPONSE: We appreciate this feedback (alongside comment #2) and amended table 2 to provide further clarification on the number of individuals without stomach contents. Rather than adding an additional row and column, the number of individuals without stomach contents is denoted in the square brackets within the total sections.
L259-260: “Number of rays without stomach contents is shown within the square brackets ([ ]) for each location and life stage.”
- Line 263: It isn't clear how you got 211 gastric lavage samples from 61 rays (only 50 with stomach contents. What have I missed?
RESPONSE: We thank the reviewer for pointing this out. The gastric lavage samples were meant to describe the subsamples of the stomach contents, or rather the number of sorted/unique tissue types within the stomach contents. As such, we have provided further clarification for the reader.
L150-153: “Subsamples of like species, hereafter referred to as gastric lavage samples, were cut into 2 x 2 mm pieces and placed into 2 mL cryovials with 70% ethanol and stored in the -20 °C freezer until DNA extraction.”
- Table 3, line 294: this table is basically a restatement of the paragraph above. Can you summarize the text in the paragraph and refer to the table?
RESPONSE: We have adjusted this paragraph to summarize Table 3.
L284-287: “Cardiid bivalves had the greatest %FO of bivalve families in all locations except Fort Pierce, whereby lucinid clams had the greatest %FO. Busycon whelks had the greatest %FO for all Atlantic coast locations, whereas tulip snails from the family Fasciolariidae had the greatest %FO for gastropods on the Gulf coast.”
L294-296: “Cardiid bivalves had the greatest %FO for either life stage considered, whereas for gastropods, tulip snails had the greatest %FO for immature rays and Busycon whelks for mature rays.”
- Figure 6, line 313: replace “the” with “each”
RESPONSE: Accepted.
L316-317: “Percentage of total body weight (using the total stomach content weight divided by the weight of each whitespotted eagle ray) with respect to whitespotted eagle ray disk width in cm.”
- Line 504-506: is this an issue? DO you recommend efforts are focused on expanding the library?
RESPONSE: Unfortunately, the limited tissue that we saved for sequencing did not extract properly, so we were unable to do so. As such, we have provided additional recommendations.
L519-522: “Future studies considering this technique should place greater emphasis on barcoding all prospective prey items, though most species encountered during opportunistic invertebrate collections had already been sequenced and were available.”
- Line 547: swap “observed” for “identified”
RESPONSE: Accepted.
L555-559: “These data suggested some dietary overlap in common prey families between the Atlantic and Gulf coasts of Florida; however, we also observed large variation in the most frequently identified families among locations on the Atlantic coast, suggesting that while whitespotted eagle rays can be considered invertivores, their diet varies substantially by location.”
- Line 548-549: this might be worth further explanation - controlling mollusc predators of commercial shellfish/shell fish restoration efforts
RESPONSE: Accepted.
L559-564: “Finally, though durophagous rays are largely thought to interact with shellfish enhancement activities in Florida, no common commercially important bivalves were observed in the stomach contents of whitespotted eagle rays, although gastropod predators of these bivalves were detected in the diet, suggesting rays can play a facilitative role at reducing predation threats on shellfish aquaculture operations.”